

# A hydro-climatological approach to predicting regional landslide probability using Landlab

Ronda Strauch[1], Erkan Istanbulluoglu[1], Sai Siddhartha Nudurupati[1], Christina Bandaragoda[1], Nicole M. Gasparini[2], and Gregory E. Tucker[3]

1. Civil and Environmental Engineering, University of Washington, Seattle, WA

2. Earth and Environmental Sciences, Tulane University, New Orleans, LA, USA

3. Cooperative Institute for Research in Environmental Sciences (CIRES) and Department of
    Geological Sciences, University of Colorado Boulder, Boulder, CO, USA

*Corresponding to:* Ronda Strauch (rstrauch@u.washington.edu)

## Abstract

We develop a hydro-climatological approach to modeling of regional shallow landslide initiation that integrates spatial and temporal dimensions of parameter uncertainty to estimate an annual probability of landslide initiation. The physically-based model couples the infinite slope stability model with a steady-state subsurface flow representation and operates on a digital elevation model. Spatially distributed raster data for soil properties and a soil evolution model

and vegetation classification from National Land Cover Data are used to derive parameters for probability distributions to represent input uncertainty. Hydrologic forcing to the model is through annual maximum recharge to subsurface flow obtained from a macroscale hydrologic model, routed on raster grid to develop subsurface flow. A Monte Carlo approach is used to generate model parameters at each grid cell and calculate probability of shallow landsliding.

We demonstrate the model in a steep mountainous region in northern Washington, U.S.A., using 30-m grid resolution over 2,700 km$^2$. The influence of soil depth on the probability of landslide initiation is investigated through comparisons among model output produced using three different soil depth scenarios reflecting uncertainty of soil depth and its potential long-term variability. We found elevation dependent patterns in probability of landslide initiation

that showed the stabilizing effects of forests in low elevations, an increased landslide probability with forest decline at mid elevations (1,400 to 2,400 m), and soil limitation and steep topographic controls at high alpine elevations and post-glacial landscapes. These dominant controls manifest in a bimodal distribution of spatial annual landslide probability. Model testing with limited observations revealed similar model confidence for the three hazard



maps, suggesting suitable use as relative hazard products. Validation of the model with observed landslides is hindered by the completeness and accuracy of the inventory, estimation of source areas, and unmapped landslides. The model is available as a component in Landlab, an open-source, Python-based landscape earth systems modeling environment, and is designed
to be easily reproduced utilizing HydroShare cyberinfrastructure.

# 1 Introduction

In steep mountainous landscapes, episodic shallow landslides (generally <2 m depth; Bordoni et al, 2015) and landslide-triggered debris flows are often the dominant form of hillside erosion and major source of sediment into streams (Benda and Dunne, 1997a, b; Goode et al., 2012).
Where landslide processes intersect with human development, they cause property damage, disruption of infrastructure, injury, and loss of life (Taylor and Brabb, 1986; Baum et al., 2008a), contribute to sedimentation in reservoirs (Bathurst et al., 2005), and may even lead to dam failures (Ghirotti, 2012). Landslides provide punctuated sediment input to streams, affecting stream geomorphology (Benda and Dunne, 1997a, 1997b) and ecosystem dynamics (Pollock,
1998; May et al., 2009).  Landslide hazard maps are a common tool used to characterize the relative potential for landslide occurrence in space, either qualitatively (using susceptibility levels) or quantitatively (using modeled landslide probabilities) (van Westen et al., 2006; Raia et al., 2014).

Our objective is to develop a parsimonious probabilistic model of shallow landslide initiation that can be implemented with minimal calibration for landslide hazard mapping using regionally available, spatially distributed input data for soil, vegetation type, local topography, and hydroclimatology. Based on the literature review presented below, we propose that a regional landslide hazard model should: (1) be flexible enough to incorporate changes in intrinsic and
extrinsic conditions, such as vegetation and climate; (2) account for spatial variability in model parameters and forcings, and (3) integrate spatial and temporal dimensions of uncertainty to quantify landslide probability.  With these principles in mind, we develop a hydro-climatological approach to modeling regional landslide hazard using the Landlab (version 1.1.0) modeling toolkit - an open-source, Python-based earth surface modeling framework that provides flexible
model customization and coupling (Hobley et al., 2017).  Next, we provide a short literature review that guides the design of our landslide modeling approach.

## 1.1 Geomorphology and Modeling Background

Landslides occur when destabilizing forces due to gravity and pore-water pressure exceed the
resisting forces of friction and cohesion over a failure plane. These forces are controlled by intrinsic hillslope conditions, including attributes of topography, such as local slope and upslope contributing area, and properties of rock, soil, and vegetation root cohesion; and extrinsic drivers of rainfall, snowmelt, and earthquakes (Crozier, 1986; Wu and Sidle, 1995; van Beek, 2002; Naudet et al., 2008). There are three primary components of a landslide: (1) a source
area or landslide scar where the initial failure begins, (2) a transmission or scour zone, such as a debris flow channel, and (3) a toe or zone of deposition (Lu and Godt, 2013).



Landslide susceptibility can be identified through numerous methods, which can be broadly grouped into empirical methods and process-based numerical models *(*Hammond et al., 1992; Wu and Sidle*,* 1995; Sidle and Ochiai, 2006).  Data-driven empirical approaches relate the
number and frequency of historical landslide observations in a region to triggering events (Caine, 1980; Crozier, 1999; Glade*,* 2001), landscape attributes (Carrara et al., 1995; Chung et al., 1995; Lee et al., 2007), or a combination of both (Kirschbaum et al., 2012) using threshold relations and various statistical models such as logistic regression, fuzzy logic, artificial neural networks, and support vector machine (Lee et al., 2007; Pardeshi et al., 2013; Chen et al.,
2014).  Empirical methods have been used for landslide susceptibility zonation or categorizing the landscape into relative landslide hazards (Sidle and Ochiai 2006).

Process-based models employ effective stress principles to characterize the destabilizing and resisting forces under hydrologic drivers (Iverson, 2000; Montrasio and Valentino 2016),
offering the ability to explore changes in environmental and climatic conditions. Such process-based models are especially useful in areas with limited landslide inventories (Pardeshi et al. 2013). Recent process-based numerical models have largely focused on improving the characterization of the space-time dynamics of subsurface flow as a driver of pore-water pressure (e.g., Baum et al., 2008b; Raia et al., 2014; Anagnostopoulos et al., 2015; Montrasio
and Valentino, 2016). Distributed hydrology models that use steady-state or transient solutions for subsurface flow depth were coupled with an infinite-slope stability model that solves the ratio of stabilizing to destabilizing forces on a failure plane parallel to the land surface (Montgomery and Dietrich, 1994; Miller, 1995; Wu and Sidle, 1995; Pack et al., 1998; Borga et al., 1998; Casadei et al., 2003; Tarolli and Tarboton, 2006; Baum et al., 2008b).
Steady-state models assume that lateral subsurface flow, driven by the topographic gradient, at each point on the landscape is in equilibrium with a steady-state recharge rate (Montgomery and Dietrich, 1994; Pack et al., 1998).  The degree of soil saturation is predicted proportional to the ratio of upslope contributing area to local slope, and a ratio of watershed recharge and
local soil transmissivity, following TOPMODEL assumptions (Beven and Kirkby, 1979; O'Loughlin, 1986; Pack et al., 1998). More recent efforts have focused on the development of transient flow models in various complexities by coupling vertical infiltration and redistribution processes in the unsaturated zone, using the Richards equation for unsaturated flow (Richards, 1931) or its variants, with lateral flow parameterizations such as kinematic wave in 1- and 2-dimensions
(Iverson, 2000; Casadei et al., 2003; Baum et al., 2008b; Godt and McKenna, 2008; Raia et al., 2014; Alvioli et al., 2014; Anagnostopoulos et al., 2015).

While transient flow models have contributed to improved understanding of the influence of weather forcing and temporal variability in precipitation on landslide initiation, they remain
tools typically applied for relatively small-scale assessments (Iverson, 2000; Raia et al. 2014). Transient models require a large number of hydrologic soil and vegetation parameters that are highly variable, uncertain, and difficult to measure or estimate (Godt and McKenna 2008; Baum et al. 2008b). In addition, in most steep forested mountains where landslide risk is high,





presence of macropores due to connected root structures, biological activity, fractures, large clasts, and lenses, leads to preferential and funneled flows that violate the assumptions of most matrix-flow models (Nimmo, 2005; Sidle et al., 2001; Gabet et al., 2003; Beven and Germann 2013). Numerical solutions to flow equations also present a major computational bottleneck in

large-scale applications for probabilistic quantification of landslide hazard.

Comparison of steady-state and transient models using case studies with known extreme rainfall events that caused widespread landsliding involve statistical model performance evaluation (Zizioli et al., 2013).  While using transient hydrologic models provided slight

improvements in the prediction of landslide locations, overall, statistical comparisons of model outputs between steady-state and transient models revealed fairly similar degrees of success (Gorsevski et al., 2006; Zizioli et al., 2013; Anagnostopoulos et al., 2015; Boroni et al., 2015; Formetta et al., 2016). In some applications, model complexity increased the accuracy of predicted landslide locations at the expense of overestimating instability on unsaturated

hillslopes (e.g., Godt et al., 2008; Bellugi 2011).  In other cases, model precision increased while accuracy decreased (Gorsevski et al., 2006).

Data uncertainty due to spatial and temporal variability of parameters continues to be one of the major challenges in predicting landslides over broad regions (Crozier, 1986; Sidle and

Ochiai, 2006; van Westen et. al., 2006; Baum et al., 2014; Anagnostopoulos et al., 2015). These uncertainties and variabilities can develop from geological anomalies, inherent spatial heterogeneities in soil and vegetation properties and their changes over time, and sampling limitations (El-Ramly et al., 2002; Cho, 2007; Baum et al., 2014).  Uncertainties in hydro-climate quantities, such as precipitation and recharge, are particularly pronounced in steep high

mountain regions due to lack of observations and complex spatial and temporal atmospheric processes (Roe, 2005; Wayland et al., 2016). Designating landslide hazard as a probability, rather than an index, systematically accounts for uncertainty and variability in stability analysis (Hammond et al., 1992) and more appropriately represents complex systems (Berti et al., 2012).  Currently, only limited process-based models account for data uncertainty in landslide

hazard mapping (e.g., Pack et al., 1998; Raia et al.,2014).

Observations and model experiments suggest that the largest landslides are usually associated with the largest rainfall events (e.g., Page et al. 1994; Gorsevski et al., 2006). Considering that hillslope hydrology is more likely to attain equilibrium conditions during prolonged wet

conditions (e.g., Barling et al., 1994; Borga et al., 2002), a steady-state representation of subsurface flow hydrology, coupled with a process-based infinite slope stability model is an efficient approach for predicting the likelihood of landslide hazard at regional scales.

Lastly, most landslide hazard methods disregard a temporal dimension over which landslide

probability is defined (Wu and Sidle, 1995; van Westen et al, 2006). As a result of that, instead of using estimated probabilities directly in the form of return periods of observed landslides or expected values for risks resulting from landslides, models use probability estimates as relative indices (eg., Pack et al., 1998) that can be used for hazard zonation (Pardeshi et al., 2013).  Lack





of temporal dimension limits the incorporation of model results into risks assessments and the decision-making processes in high-risk regions.

## 1.2 Approach Overview

We develop a process-based modeling approach for shallow landslide initiation that incorporates imprecisions and uncertainties in hydro-climatological forcing, soils, and land cover properties. Rather than predicting critical rainfall intensity necessary to destabilize hillslopes (Montgomery and Dietrich 1994) or a terrain stability index map (Pack et al. 2001, 2005), our approach aims to develop a spatially continuous probability of landslide *initiation*

that can be updated as conditions and triggers evolve. The model evaluates the infinite slope stability equation at the scale of a grid cell from a Digital Elevation Model (DEM). Daily rate of recharge (i.e., flux of water entering saturated zone) can be provided by model users from a variety of grid resolutions from hydrologic models such as the Variable Infiltration Capacity (VIC) model (*Liang et al.* 1994) as used in our regional application, or assigned as parameters by

the user. A "Source Tracking Algorithm" (STA) is developed to route spatially variable recharge fields, at the native resolution of a hydrology model, generically referred to as a Hydrology Source Domain (HSD), onto the grid resolution of slope stability calculations. Raster grids derived from soil texture and vegetation cover classes are used with look-up tables to estimate model parameters ranges obtained from the literature to quantify uncertainty. Through Monte

Carlo simulation (Raia et al., 2014), we calculate the probability of landslide initiation at each landscape grid cell. Our probability is further refined by a geomorphic soil evolution model that estimates soil depth with greater spatial heterogeneity than conventional soil survey map units, which is critical for slope stability analysis (Dietrich et al., 1995). This soil evolution model estimates long-term soil depth based primarily on soil mass production and slope-dependent

sediment transport rules.

Landslide probability calculations are written in Python as a Landlab LandslideProbability component ([landlab.github.io](landlab.github.io), including User Manual). The STA is available as a Landlab utility. The landslide model is designed as a user-written "driver" within Jupyter Notebooks, where the

workflow of model application is presented. The driver and data are deployed on HydroShare ([www.hydroshare.org](www.hydroshare.org)), an online collaboration environment for sharing data, models, and code (Horsburgh et al., 2016; Idaszak et al., 2016), and made available for cloud computing via HydroShare JupyterHub infrastructure using a web browser (see Sect. 2.5).

In this work we explore the questions (1) How does regional hydro-climatology influence the spatial patterns of shallow landslide initiation over large geographic scales? and (2) How does distributed soil depth influence the probabilistic nature of landslide initiation compared to coarse-scale, homogenous soil depth estimates? We demonstrate our approach in a mountainous region of Washington, USA. This Pacific Northwest (PNW) region is naturally

susceptible to landslides because of high and intense rainfall, steep mountains, active tectonics, and geologic and glacial history (Nadim et al., 2006; Sidle and Ochiai, 2006). The Oso landslide, which occurred in the vicinity of our study area in 2014, resulting in 43 fatalities and over $50 million in economic losses, provides a solemn reminder of the hazard landslides present





Earth **Surface**
Dynamics
Discussions

(Wartman et al., 2016). Although the Oso landslide was a deep-seated type, the greater frequency of shallow landslides affords utility and relevance to our model.

## 2 Methodology

### 2.1 Probabilistic approach to landslide initiation

Our approach is based on the infinite slope stability equation derived from the Mohr-Coulomb failure law that predicts the factor-of-safety (FS) stability index of a hillslope parcel from the ratio of stabilizing forces of soil cohesion and friction, reduced by pore-water pressure of subsurface flow, to destabilizing forces of gravity (Hammond et al., 1992; Wu and Sidle, 1995). The model as given by Pack et al. (1998) is:

$$FS = \frac{(C_r + C_S)/h_s \rho_s g}{\sin \theta} + \frac{\cos \theta \tan \phi (1 - R_w \rho_w / \rho_s)}{\sin \theta} \qquad (1a)$$

$$C* = (C_r + C_s)/h_s \rho_s g \qquad (1b)$$

C* is a dimensionless cohesion (Eq. 1b) embodying the relative contribution of cohesive forces to slope stability. When C*>1, cohesion is sufficient to hold the soil slab vertically (Pack et al., 1998). $Cr$ and $Cs$ are root and soil cohesion respectively [Pa], $h_s$ is the soil depth perpendicular
to slope [m], $\rho_s$ and $\rho_w$ are saturated soil bulk density and water density [kg/m$^3$], respectively, $g$ is acceleration due to gravity [m/s$^2$], $\theta$ is slope angle of the ground, and $\phi$ is soil internal friction angle [°]. Relative wetness, $R_w$, is defined as the ratio of subsurface flow depth, $h_w$, flowing parallel to the soil surface, to $h_s$. Deterministically, a hillslope element is unstable if $FS < 1$ and stable if $FS > 1$ (Sidle and Ochiai, 2006; Shelby, 1993). When FS = 1, the slope is "just-stable" or
in a state of "limited equilibrium" (Lu and Godt, 2013).

Relative wetness is arguably the most dynamic factor at short time scales, relating to water table depth and to recharge rate. It is derived from local subsurface lateral flow, q$_s$ [m$^2$ d$^{-1}$], represented by a 1-D (i.e., flow parallel to bedrock) form of the kinematic wave approximated
by Darcy's law using topographic gradient of hillslope, q$_s$=K$_s$h$_w$sin$\theta$ (Wu and Sidle, 1995). Under a steady-state assumption, lateral flow is in balance with the rate of water input, q$_r$ [m$^2$ d$^{-1}$], through a uniform rate of recharge, R [m d$^{-1}$], across the upslope specific contributing area, $a$ [m], q$_r$=Ra. This assumption gives: Ra=K$_s$h$_w$sin$\theta$, where K$_s$ is saturated hydraulic conductivity [m d$^{-1}$]. Solving this equation for h$_w$ and dividing both sides by h$_s$ gives R$_w$ (Montgomery and
Dietrich, 1994; Pack et al., 1998):

$$R_w = \frac{h_w}{h_s} = \min\left(\frac{R\,a}{T\,\sin\theta}, 1\right) \qquad (2)$$

Here $T$ is soil transmissivity [m$^2$ d$^{-1}$], which is depth-integrated saturated hydraulic conductivity, K$_s$. For uniform $K_s$ within the soil profile, $T=K_s h_s$. Ground saturates when $R_w = 1$, which represents hydrostatic conditions and the maximum value for $R_w$. Options for user-provided $T$
or $K_s$ are accepted by the component; although comparison of resulting probabilities were found to be similar given that the value of $T$ was derived from $h_s$. We assume uniform conductivity within the soil profile overlying a relatively impermeable layer such as bedrock,





and subsurface flow direction parallel to this drainage barrier (Montgomery and Dietrich, 1994). These assumptions are appropriate for relatively steep topography and to efficiently characterize wetness over large areas (Tarolli and Tarboton, 2006; van Westen et. al., 2006).

A Monte Carlo simulation is used with equation (1) by assuming $R$, $T$, $C$ ($C=C_r+C_s$), $h_s$ and $\phi$ as random variables represented by probability distributions (Tobutt, 1982; Hammond et al., 1992). One benefit of Monte Carlo simulation is that many of the sources of inaccuracy (e.g., nonlinearity, input uncertainties) are overcome (Strenk, 2010; El-Ramly et al., 2002) by generating a distribution of samples over a plausible range for selected variables.  The
uncertainty in R is defined by using a time series of the maximum daily recharge in each year (e.g., Benda and Dunne, 1997a; Borga et al., 2002; Istanbulluoglu et al., 2004). The model includes both spatially uniform and spatially distributed options for sampling recharge (described further in Sect. 2.3).  Using sampled random variables in Eq. (1a), FS is calculated in each model iteration during the simulation. Annual probability of failure P(F) and landslide
return period (RP) at each grid cell are defined as (Hammond et al., 1992; Cullen and Frey, 1999):

$$P(F) = \ P(FS \ \leq 1) = \ n(FS \leq 1)/n \qquad (3a)$$
$$RP = P(F)^{-1} \qquad (3b)$$

Our model does not predict the size of a probable landslide at the initiation point, which can be
smaller or larger than the size of a DEM grid. P(F) gives a relative propensity that a landslide could initiate within the grid cell.  The design of the model reflects the uncertainty of soil and vegetation within a grid cell.  Therefore, if some random samples lead to a low deterministic FS, they contribute to an increase of the P(F) within that cell.

## 2.2 Model Development in Landlab
The landslide modeling approach presented above is implemented in Landlab (landlab.github.io). Landlab is an open-source modeling toolkit written in Python for building and running two-dimensional numerical models of Earth-surface dynamics (Tucker et al., 2016; Hobley et al., 2017; Adams et al., 2017). A detailed explanation of the Landlab framework is
provided in Hobley et al. (2017). Landlab provides a grid architecture, a suite of pre-built components for modeling surface or near-surface processes, and utilities that handle data creation, management, and interoperability among process components. The Landlab design allows for a "plug-and-play" style of model development, where process "components" can be coupled together in a user-customized "model driver".  Each component is a set of code
functions that represent an individual process; the model driver has code used to import or generate required data, execute the component or set of components used in the model, and to visualize results.  For example, once a DEM is imported as a Landlab grid instance, any Landlab component can be used with interoperable methods to attach data and perform operations. Landlab code developed for this work is explained in detail in the user manual of
the Landlab LandslideProbability component available from eSurf and the Landlab github website.



The Landlab workflow developed in this regional landslide probability mapping study uses the LandslideProbability component presented in Fig. 1. The workflow includes preparing spatial model parameters and model forcing data completed in preprocessing steps outside of

Landlab. A model driver is written to run the LandslideProbability component on RasterModelGrid (RMG) instance only. RMG is a Landlab class for creating raster grids and representing the connections among grid elements. A structured grid is generated that covers the model domain. Spatial model parameters and forcing variables supplied by the user are stored on the grid elements as Landlab data fields, which are NumPy arrays containing data

associated with grid elements (in this case nodes). The driver imports Landlab and necessary Python libraries as well as loads and processes data required for the LandslideProbability component.

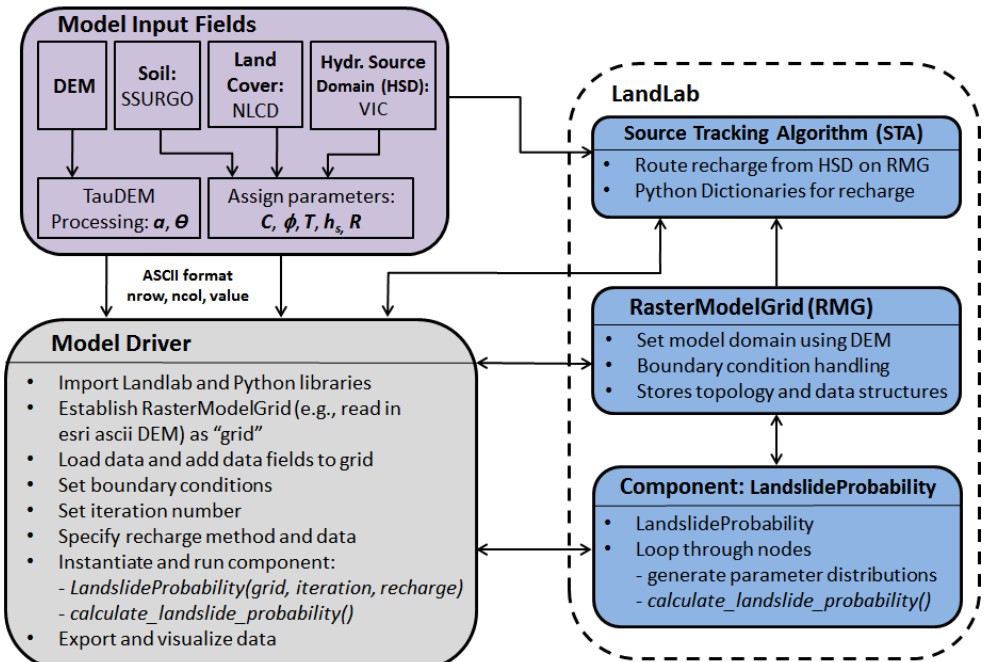

**Figure 1.** Workflow for landslide modeling using the Landlab LandslideProbability component. The user
creates input parameter fields (purple box). The model driver (gray) imports Landlab, Python libraries, and model parameters fields: instantiates (e.g., create an instance) the RasterModelGrid and the component; and runs utilities and methods of Landlab (blue inside dashed box).

Slope angle and specific contributing area are static parameters derived from a DEM in pre-processing steps. Total cohesion, $C$ (i.e., $C_r + C_s$), $\phi$, $h_s$, and $T$ are treated as random variables
following a triangular distribution specified with three parameters (minimum, mode, and maximum) to represent spatial and temporal uncertainties in these parameters on the landscape (Cho, 2007; Dou et al., 2014). Triangular distributions give weight to the most likely value (i.e., mode) and have been proposed in other Monte Carlo simulations of slope stability

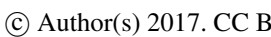


(Hammond et al., 1992; El-Ramly et al., 2002; Strenk, 2010). Parameters of the triangular distribution can be assigned by relating categorical vegetation cover variables, for example from the National Land Cover Data (NLCD) (Jin, 2013; USGS, 2014b) or other map sources, with a lookup table for cohesion and using available soils data such as gridded Soil Survey
Geographic Database (SSURGO) (*DOA-NRCS* 2016), to assign internal friction angle, soil depth, and transmissivity (see 3.3 for details). Soil density is set as a constant field, 2,000 kg m$^{-3}$ in our application.

In each Monte Carlo iteration, we characterize recharge as an annual maximum daily recharge
event. Four options for sampling recharge are provided, which are identified in the model driver by selecting a probability *distribution*: *uniform, lognormal, lognormal_spatial,* and *data_driven_spatial*. The first two assign spatially uniform random variables with respective parameters of minimum and maximum, and mean and standard deviation. The latter two are designed to represent spatial variability in recharge based on historical annual maximum daily
recharge routed to each node of the model domain. The *lognormal_spatial* option assigns mean and standard deviation of annual maximum recharge and uses lognormal distribution of recharge for simulation. The *data_driven_spatial* option uses a non-parametric Monte Carlo sampling approach to sample directly from historical recharge data. Upslope-averaged recharge for each grid node is calculated with the Landlab Source Tracking Algorithm (STA) utility using
recharge from a HSD, which in this study is the VIC macroscale (1/16° or 5x6 km grid cell) hydrology model.

Within the model driver, the user also sets any boundary conditions, such as areas to exclude (i.e., bedrock outcrops, glaciers) and assigning the number of Monte Carlo iterations (n>>1,000,
Hammond et al., 1992).  The LandslideProbability component is instantiated by passing four arguments: the grid, number of iterations, recharge distribution, and recharge parameters. Multiple instances of the LandslideProbability class can be established in one driver to compare the results from different recharge specifications.  Once the component has been instantiated, the component's method *calculate_landslide_probability()* is run.  For each iteration, this
method loops through each core node, generates unique model parameters, and calculates the relative wetness (Eq. 2) and deterministic FS index (Eq. 1a) at each iteration. At the end of the iterations, the P(F) at the node is calculated as the number of iterations in which FS≤1 divided by the number of iterations.  Variables output by the component at each core node include calculated probability of saturation and P(F), which can be queried at each node or visualized
across the entire grid within the driver or using a command line terminal to execute commands.

## 2.3 Hydrologic Data Processing
A key aspect of the regional landslide modeling approach is the linking of landslide hazard to hydro-climatological forcing at regional scales. The Landlab LandslideProbabilty component is
written with the capability to accept input from hydrologic model outputs.  We used the VIC macroscale hydrologic model (Liang et al., 1994) to demonstrate this capability because it characterizes elevation-dependent differences in regional precipitation and temperature forcings and their influence on recharge through regulating snow accumulation and melt, rain-



on-snow, evapotranspiration, and soil moisture. VIC is semi-distributed, predominantly physics-based macro-scale hydrology model, which is advantageous for representing distributed parameters of hydro-climatology that are not stationary in time over large regional areas (Hamlet et al., 2013).

The VIC model simulates the land surface as a large, flat, uniform grid with sub-grid heterogeneity (e.g., vegetation and elevation) based on statistical distributions. Daily or sub-daily meteorological drivers (e.g., temperature and precipitation) influence the fluxes of water and energy near the land surface. Each grid is simulated independently and flows between grid

cells are ignored (e.g., unrouted). Precipitation enters the upper of typically three layers of soil and infiltrates to lower layers via a variable infiltration curve. Soil water can move between layers vertically and is lost through evapotranspiration and from the third layer as base/subsurface flow via non-linear recession. Water input in excess of infiltration forms surface runoff.

To characterize the annual probability when the ground is likely to be the most saturated, daily baseflow and surface runoff are summed at each VIC grid cell to represent recharge [mm d$^{-1}$] and the annual maximum daily value is selected for each model year, similar to others (e.g., Benda and Dunne, 1997a; Borga et al., 2002; Istanbulluoglu et al., 2004). The recharge data

arrays are keyed to latitude, longitude, and grid cell ID (a user-defined ID for each VIC grid cell, in our case) packaged as Python dictionaries (see Fig 1. of User Manual). To help account for lateral fluxes in groundwater (van Beek, 2002), VIC recharge is routed to each node in the model grid using the STA utility, which also addresses the different spatial resolutions of VIC and the RMG. This Landlab utility was developed to derive the fraction of annual maximum

recharge from each VIC grid cell within the upslope contributing area of each Landlab grid node. The fractions and VIC IDs are saved as values for two Python dictionaries keyed to the RMG node ID. At each node, these dictionaries are used to calculate the upstream proportionally-averaged maximum recharge for each year.

**2.4 Soil Depth Evolution Model**
Soil depth controls the temporal and spatial patterns of landsliding over geomorphic time scales and is considered one of the most significant parameters controlling the FS stability index, especially at depths less than 1.5 m (Benda and Dunne, 1997a; Istanbulluoglu et al., 2004; Catani et al., 2010; Sidle and Ochiai, 2006). Soil depth can vary in space and time as a function

of weathering and sediment transport in relation to climate, lithology, topographic position, and vegetation cover (Dietrich et al., 1995). As an alternative to spatial soil maps such as the SSURGO database (*DOA-NRCS* 2016), which are often produced at the soil pedon-level, we developed a soil depth map using a simple soil evolution model and topographic and land cover attributes (Dietrich et al., 1995; Pelletier and Rasmussen, 2009; Tesfa et al., 2009; Bellugi et al.,

40    2015).




Change in soil depth depends on soil production by bedrock weathering and slope-dependent sediment transport expressed as (Nicótina et al., 2011; Tucker and Slingerland, 1997; Heimsath et al., 1997):

$$\rho_s \frac{\partial h_s}{\partial t} = -\rho_r \frac{\partial z_b}{\partial t} - \rho_s \nabla q_s \tag{4}$$

5    where $\rho_s$ and $\rho_r$ are bulk densities for soil and rock, respectively, $h_s$ is soil depth, $z_b$ is the elevation of the soil-bedrock interface, $t$ is time, $\nabla$ is the topographic divergence operator for the topographic gradient, and $q_s$ is the sediment flux. The soil production rate (e.g., first term in Eq. 4 on right side) is a function of the rate of change in the elevation of the soil-bedrock interface, which has been shown to decline exponentially with soil depth (Heimsath et al., 1997, Gabet et al., 2003):

$$\frac{\partial z_b}{\partial t} = -P_o e^{-\alpha h_s} \tag{5}$$

where $P_o$ is the soil production rate from exposed bedrock (i.e., no soil cover) and $\alpha$ is the rate of exponential decay with depth. Diffusive sediment transport characterized in the second term on the right side of Eq. (4) can be represented by a simple soil creep function dominant in convex hillslopes as (Nicótina et al., 2011; Istanbulluoglu et al., 2004):

$$\nabla q_s = -K_d \nabla^2 z \tag{6}$$

where $K_d$ is a linear hillslope diffusion coefficient and $\nabla^2$ is Laplacian of elevation. Dividing Eq. (4) by $\rho_r$, multiplying by the ratio of $\rho_r / \rho_s$, and substituting Eq. (5) and Eq. (6) into Eq. (4), yields the following instantaneous soil depth equation:

$$\frac{\partial h_s}{\partial t} = \frac{\rho_r}{\rho_s} P_o e^{-\alpha h_s} + K_d \nabla^2 z \tag{7}$$

The change in soil depth with time based on Eq. (7) is added to the soil depth at *t-1* (*t* in years) to evolve soil depth over time (see also: Pelletier and Rasmussen 2009).

Variable curvature profiles, steep and planar hillslopes, and abrupt knife edge drainage divides indicate nonlinear transport processes such as mass wasting (Roering et al., 2004, 1999). These landscape characteristics are common in the steep terrain; therefore, in every iteration of the model Eq. (a) and Eq. (2) are used to calculate FS within the soil evolution model. When FS≤1, soil is removed to bedrock by setting it to a very small value of 0.005 m to be consistent with the creep equation. In each model iteration, C and T, were randomly sampled and used deterministically in the FS Eq. (1a). Calibration of the soil evolution model is done by adjusting $P_o$ and $K_d$ for the location of the landslide analysis based on published long-term rates of erosion and diffusion. Creation, calibration, and application of the soil evolution model are detailed in Sect. 4.1.2.

## 2.5 Reproducibility

To publish a reproducible version of this research, we used the HydroShare (www.hydroshare.org) cyberinfrastructure platform, which is designed explicitly to encourage the reusing and sharing of models (Tarboton et al., 2014; Horsburgh et al., 2016; Morsy et al., 2017). Steps that supported reproducibility included using the HydroShare sharing settings





with a workflow that started with *Private* while data and models were developed, *Discoverable* while research was being shared with colleagues for review, and *Public*, once our results were accepted for publication. We used the *Select a license* function to add No Commercial (NC) use to our Creative Commons license. We made use of the *Groups* social collaboration, by making

early versions of our research results available to invited participants of workshops and tutorial demonstrations to our Landlab group in HydroShare. The data and model are accessed by launching Jupyter Notebooks that access Landlab installed on JupyterHub servers at the National Center for Supercomputing Applications (Yin et al, 2017; Castranova, 2017). HydroShare features enable our current and future researchers to use the *Copy Resource*

function to replicate our published resource (i.e., the landslide model) in their own account with *Derived from* metadata that references back to the published resource DOI, to serve as a starting point for their work.

## 3 Model application

### 3.1 Study Area

The model described above is applied within the geographical limits of the North Cascades National Park Complex (NOCA) in the state of Washington, U.S.A, managed by the U.S. National Park Service (Fig. 2). In recent decades, NOCA has experienced damaging and disruptive landslides that have impacted infrastructure and the public. Furthermore, the park area is covered by a recent soil survey between 2003 and 2009, including field investigation (DOA-

NRCS and DOI-NPS, 2012), and has a complete map of mass wasting processes visually observed in the field (Riedel and Probala, 2005).

NOCA is approximately 2,757 km$^2$, with 93% wilderness (in which no motorized or mechanized devices are permitted; DOI-NPS, 2012), which is ideal for studying naturally triggered landslides.

The elevation ranges from about 100 m to 2,800 m (Fig. 2a). The terrain is composed of rock slopes at the highest elevations, short (<100 m) soil-mantled hillslopes, and landslides upslope of relatively straight debris flow channels connected to the fluvial system. Over 300 glaciers occupy mountain peaks in NOCA. The influence of the Pacific Ocean, approximately 80 km to the west, provides a humid temperate climate. However, the north-south oriented Cascade

Mountains create an effective orographic climate boundary, separating a wetter west side from a drier east side. Reported mean annual precipitation ranges from about 708 mm in the low elevations of the eastern slopes to 4,575 mm at the highest mountain elevations west of the Cascade crest, with about 70% falling in November through March (Fig. 2b). This spatial precipitation gradient is the result of orographically-enhanced precipitation that leads to a

strong rain shadow (Roe 2005). Average annual air temperatures range from -2 to 11°C, depending on elevation (DOA-NRCS and DOI-NPS, 2012).

Vegetation is mainly coniferous trees, with deciduous trees along river floodplains, and shrubs, meadows, and barren land in the subalpine and alpine environments. In this study vegetation

classes were grouped into herbaceous, shrubland, and forest using the 2014 NLCD data, which is based on the land use/land cover (LULC) classification of 2011 Landsat satellite imagery (Jin,





2013; USGS, 2014b).  Based on this classification, forest, shrubs, and herbaceous vegetation represent 58%, 17%, and 12% of the park, respectively. Elevation ranges for these vegetation classes are from 106 to 2363 m (forest), 110 to 2465 m (shrubs), and 121 to 2759 m (herbaceous).

The park geology is composed of a complex mosaic that includes mostly complexly faulted and folded sedimentary and volcanic rocks on the west side, unmetamorphosed sedimentary and volcanic rock on the eastern edge, and highly squeezed and recrystallized metamorphic rock originating from great depth in middle (Haugerud and Tabor, 2009).  Alpine and continental

10    glaciation, along with rivers and mass-wasting processes, have created the landscape we observe today. The glaciers eroded U-shaped valleys with steep valley walls prone to landslides and flat valley floors with gravel-bed rivers. The lower ends of many valleys on the east side were not covered in alpine glaciers and have narrow, winding V-shaped canyons and steep, narrow rivers.

A park-wide landform mapping study identified six different types of mass wasting: rock fall/topple, debris avalanche, debris torrent, slump/creep, sackung, and snow avalanche-impacted landforms (Riedel et al., 2015). Mass wasting landforms were identified in the landform mapping using 1998 air photos at 1:12,000 scale, 7.5 minute topographic maps,

20    bedrock geology maps, and field investigations.  The minimum mapping unit was approximately 1,000 m$^2$, except for a few smaller slump units. In this study, we only used mapped debris avalanches for model confirmation as they often initiate by shallow landslide processes. Debris avalanches typically represent a mixture of failed rock and debris and the mapped polygon included head scar, transport and scour channels, and deposition zone represented in a single

25    polygon (Fig 3a).  We extract the highest 10% of the elevations in the mapped debris avalanche polygons as landslide source areas through comparison to aerial imagery (Tarolli and Tarboton, 2006). Landslide sources are more frequent in the intermediate elevations. In the NOCA region, 75% of landslide source areas are located in the 1,200 m to 2,000 m elevation range (Fig. 3b).





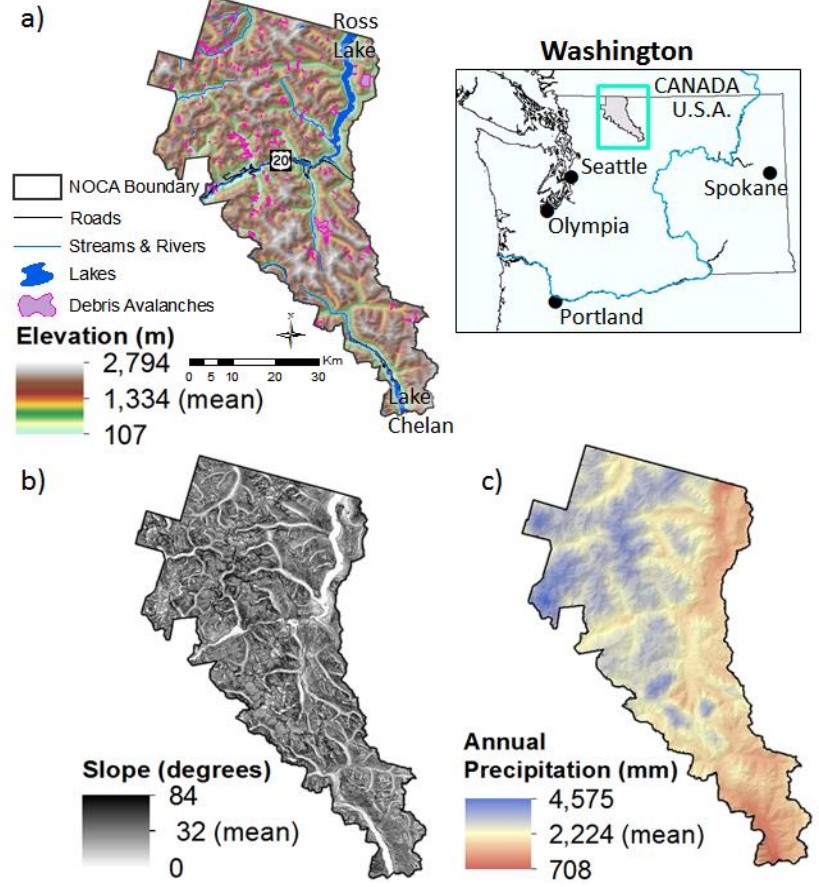

**Figure 2.** North Cascades National Park Complex (NOCA) in northern Washington state, U.S.A:
**(a)** a 30-m DEM of the domain overlain by debris avalanches and major water bodies; **(b)** slope derived
from DEM; and **(c)** mean annual precipitation (1981-2010 average) mapped at 800-m resolution from
PRISM (PRISM Climate Group, 2004).

Some areas in mountainous regions are covered by glaciers, permanent snowfields, and
exposed bedrock, which are unsuitable locations to model landslides on soil-mantled hillslopes
using the infinite slope model (Borga et al., 2002). Furthermore, they are not expected to be
destabilized by precipitation, although other forces could cause failures (e.g., earthquake,
volcanic activity, and temperature). We exclude high elevation areas covered by glaciers,
permanent snowfields and exposed bedrock (Fig 3c), as well as wetlands and other water
surfaces, based on landform mapping and maps of lithology and LULC, from our modeling
domain and geomorphic analysis because shallow landslides are not typically observed on these
landforms. The total area excluded from the stability analysis accounts for about 21% of NOCA's
land area.





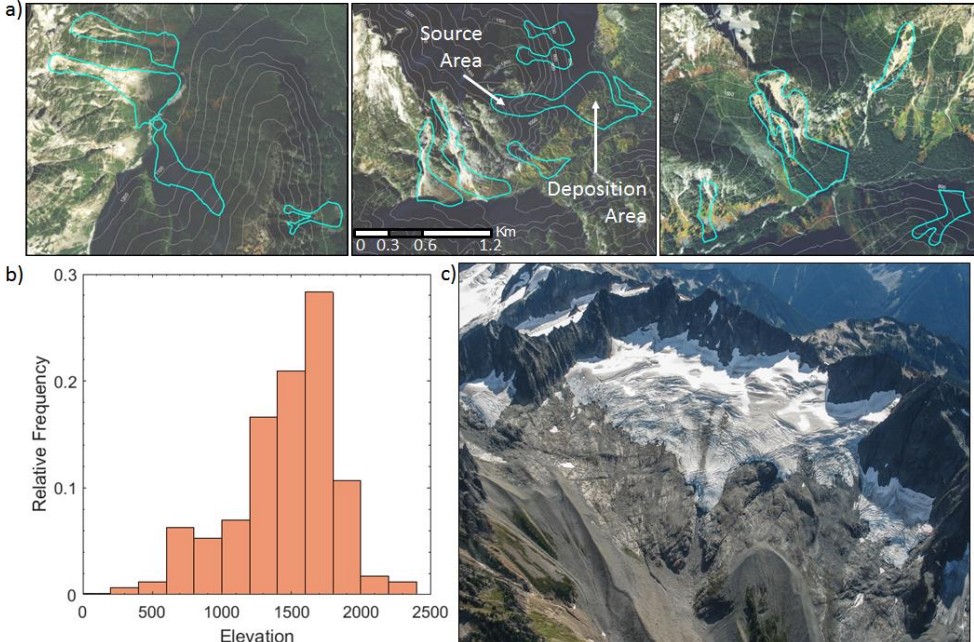

**Figure 3. (a)** Example debris avalanches (cyan) mapped in three areas within NOCA. Contours are in 100-m intervals. Aerial image source from World Imagery, Esri Inc.[1]; **(b)** elevation distribution of the relative frequency of mapped debris avalanche source areas (upper 10%); and **(c)** High elevation rock and glacier mapped surrounding Spiral Glacier in North Cascades showing a bedrock glacier cirque with thin barren soils and moraine deposits (photo by John Scurlock with permission).

### 3.2 Model Input Fields

We used a grid resolution of 30 m to evaluate and compare our regional model of landslide probability to a limited set of landslide observations. A 30-m grid cell size is consistent with the minimum mapping unit used for landslides (Riedel et al., 2015; see also Regmi et al., 2014). Slope (S=$tan\theta$), combined curvature (Curv), and contributing area (CA) attributes were derived from a 30-m DEM acquired from *National Elevation Dataset* (NED) (USGS, 2014a) (Fig. 2a). In addition, the NLCD data for vegetation classification and the SSURGO soils database we used in this study both have available 30-m grid resolutions. To show the model potential for regional applications, a global coverage of 30-m DEM from the NASA Shuttle Radar Topography Mission (SRTM) is available (USGS 2017). Thus, showing the model's potential at this resolution is intended in this paper, especially for regional applications beyond the use in a single watershed across the globe.

---

[1] Images created using ArcGIS® software by Esri. ArcGIS® and ArcMap™ are the intellectual property of Esri and are used herein under license. Copyright © Esri. All rights reserved. For more information about Esri® software, please visit www.esri.com.



### 3.2.1. Vegetation and Soil parameters

Parameters of a triangular distribution for *C, ø, T, and $h_s$* are provided in Table 1. In our case study, *C* represents root cohesion because we assumed soils to be primarily cohesionless, due to low clay content in this mountain substrate. We developed spatial coverages for minimum, mode, and maximum C for NOCA by relating vegetation classes with corresponding published C values in the literature (Table 1), where field observations suggest right-skewed distribution (Hammond et al., 1992; Schmidt et al., 2001; Gabet and Dunne 2002; Hales et al. 2013). Based on ranges available in the literature, we selected a mode value as a commonly reported value, minimum parameter as 30% of the mode, representing death and loss of productivity (Sidle, 1991; 1992), and a maximum near the highest reported value for C. Other LULC types include water, wetland, snow/ice, barren, and developed (e.g., roads, campground). Small C values are assigned for barren and developed land uses (~14% of the domain) having minimal vegetation. Mode values of C mapped over NOCA are given in Fig. 4b.

**Table 1.** Parameters defined for vegetation and soil types in the study region. For spatially continuous parameters, values represent the statistics for the model domain with *(mean)* values in parentheses.

| Parameter | Minimum | Mode *(Mean)* | Maximum |
|---|---|---|---|
| **Root Cohesion [kPa]** | | | |
| Barren/Developed | 0.03 | 0.10 | 0.15 |
| Forest (coniferous) | 3 | 10 | 20 |
| Shrubland | 1.2 | 4 | 10 |
| Herbaceous | 0.6 | 2 | 5 |
| **Internal angle of friction [°][1]** | | | |
| Loamy sand | 26.2 | 32 | 42.2 |
| Sandy loam | 28.7 | 35 | 46.2 |
| Developed areas (loamy, sandy) | 28.7, 31.2 | 35, 38 | 46.2, 50.2 |
| **Transmissivity [m² d⁻¹] [2]** | 0.42 | *(3.39)* | 16.4 |
| **Soil depth [m] †** | 0.09 | *(0.62)* | 2.01 |

[1] Developed areas within the two soil types, respectively, have mode values 3° larger due to compaction.
[2] Values for the continuous variables, transmissivity and soil depth, represent the minimum, mean, and maximum for the study area, not individual soil map units.

Despite the aggregation of plant types into functional plant communities (Fig. 4a), considerable spatial variability in C is present within the park (Fig. 4b), with the greatest values in the forest communities of the valley bottom and lower valley walls. As communities transition from forest to shrublands to herbaceous species with increasing elevation, C declines. Note that herbaceous species are likely composed of considerable woody vegetation in this alpine region, but of diminutive stature.

In order to investigate the contribution of soil depth to mapping landslide probability, we developed and used two alternative soil depth products. The nationally available SSURGO



database maintained by the Natural Resources Conservation Service (NRCS) is a readily available data source that includes depth-to-restrictive layer (*DOA-NRCS* 2016), which we used to specify the mode of soil depth (Fig. 4c). Using the *Soil Data Viewer* of Esri ArcGIS (DOA-NRCS, 2015a), the weighted-average aggregation option is used to extract soil depth within each soil

map unit (DOA-NRCS and DOI-NPS, 2012). SSURGO soil depth (SSURGO-SD) is uniform for each soil map unit and thus, lacks finer scale spatial heterogeneity and create edge incongruities (Fig. 4c), a limitation also identified in other landslide modeling studies (Bordoni et al., 2015). A smoother and spatially consistent soil depth map is achieved using the soil evolution model.

SSURGO-SD represents the recent conditions in soil depth. The difference between the actual soil depth in the field and the SSURGO reported soil depth will likely be associated with the limited number of soil depth measurements used to develop SSURGO maps, measurement errors, and spatial interpolation assumptions. In addition, for the locations that have already produced landslides before SSURGO mapping, we assume that the maximum value of the

triangular distribution represents the soil depth prior to a landslide. To represent uncertainty, minimum $h_s$ is assumed to be 70% of the mode and maximum $h_s$ adds 10% to the mode value. These values give a left-skewed triangular distribution, commonly used in probabilistic landslide models (Hammond et. al., 1992). Selected ranges were confirmed by the soil evolution model discussed in Sect. 4.1.2.

Transmissivity is derived as the product of weighted-average aggregated $K_s$ of all soil layers above the restrictive layer and $h_s$ for each soil map unit (DOA-NRCS, 2015a). Similar to $h_s$, this $T$ value was considered the mode (Fig. 4d) and the minimum and maximum values needed for an asymmetrical triangular distribution calculated as: $T_{min} = T_{mode} - 0.3*T_{mode}$ and $T_{max} = T_{mode} +$

$0.1*T_{mode}$. Closely related to soil depth, transmissivity is high in valley bottoms as well on plateaus because of deeper soils, thus, more water can move through the soil when saturated (Fig. 4d). Transmissivity is low in the thin veneer soils below retreating glaciers as well on steeper side slopes.

Soil surface texture is a grouping used to describe the particle size distribution of granular media, and can be used as an indicator of ∅ (Nimmo, 2005). The percent sand, silt, and clay (weighted-average aggregation) for each soil map unit in NOCA were derived from SSURGO data using Soil Data Viewer (DOA-NRCS, 2015b). This revealed largely sandy loam or loamy sand soil textures, based on USDA classification, across the NOCA. These soil textures corresponded

to Unified Soil Classification System (USCS) soil types silty sand and well-graded (diverse particle size) fine to coarse sand, respectively. Reported ∅ values for these USCS soil types were assigned as the $∅_{mode}$ (i.e., Table 5.5 in Hammond et al., 1992 and Table 5.2 in Shelby, 1993). Developed land cover type was assigned an additional 3° to the mode to compensate for higher soil density from development activity, such as compaction (Sidle and Ochiai, 2006). The map of

∅ exhibits the least variability in NOCA due to the relatively narrow range of soil textures, with lower angles typical at higher elevation and higher angles farther downslope (figure not shown). Given the mode and ranges of ∅ for these soil types, minimum and maximum ∅ were calculated to generate right-skewed distributions for both soil types as: $∅_{min} = ∅_{mode} - 0.18*∅_{mode}$




and $\varnothing_{max} = \varnothing_{mode} + 0.32*\varnothing_{mode}$. The soil and water density terms in Eq. (1a), were assigned a constant value of 2,000 kg m$^{-3}$ and 1,000 kg m$^{-3}$, respectively (Pack et al., 2005).

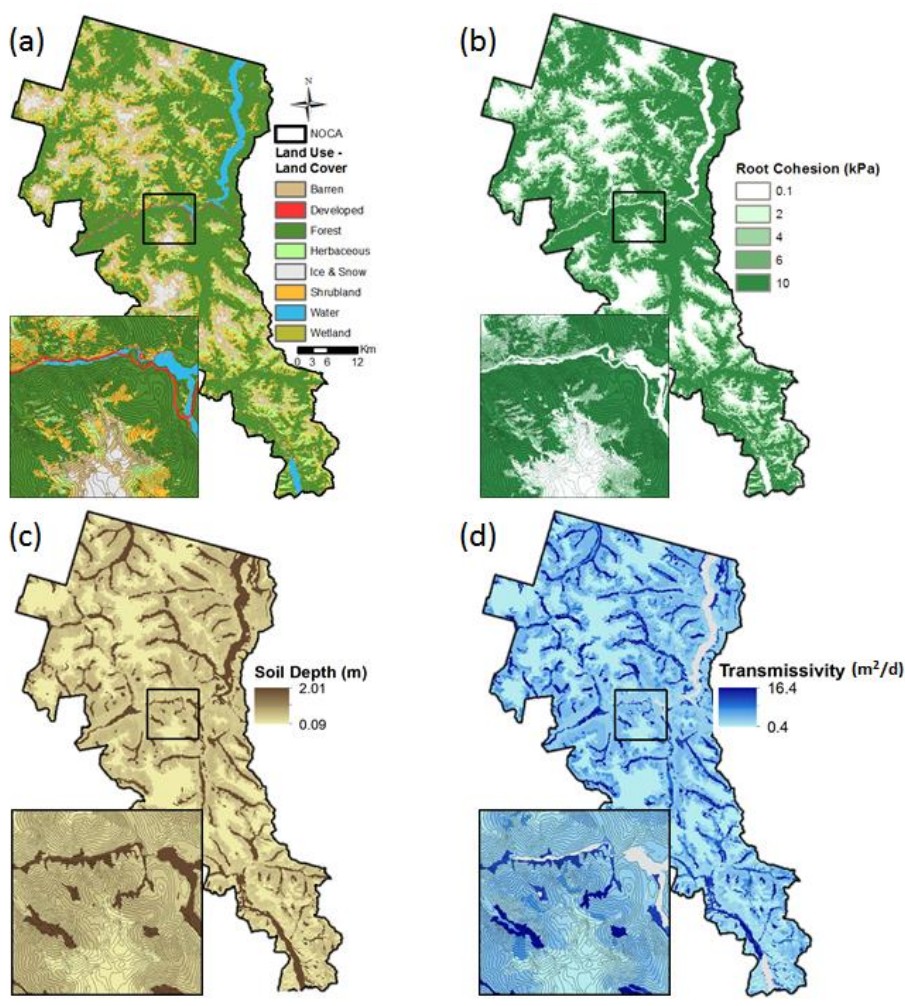

**Figure 4.** Distributed parameters used in the landslide model over NOCA, including: **(a)** LULC classified
10  from NLCD (2014); **(b)** root cohesion based on LULC; **(c)** soil depth from SSURGO; and **(d)** transmissivity
based on SSURGO soil depth.  Mapped values in (b) through (d) represent the mode values used in the
parameter distributions. Insert shows zoomed-in area with 100 m contours.



### 3.2.2. Model Recharge

The model is designed with a flexible approach to parameterizing recharge. Available probability distributions include uniform, lognormal, lognormal spatial, and data driven spatial. Supplemental materials include a Jupyter Notebook that reproduces these four recharge

options on a synthetic grid.  To provide the hydro-climatology forcing to drive our landslide model, our model application leverages the existing detailed simulations of VIC in the PNW region developed through the *Columbia Basin Climate Change Scenarios Project* (Elsner et al., 2010; Hamlet et al., 2013).  The project developed a calibrated implementation of VIC (1/16° or 5x7 km grid resolution) covering the Columbia River basin in Washington to produce validated

historical hydrologic simulations (water years 1916-2006) driven by spatially interpolated daily station observations of temperature and precipitation (Hamlet et al., 2013). Archived model output at a daily-time-step includes gridded baseflow and runoff.  Hydrologic simulations using VIC have also been run for all of the contiguous United States (CONUS; Livneh et al., 2013, 2015).  Thus, acquisition of hydrologic model output is readily available to apply the landslide

model anywhere throughout the CONUS. We determined the maximum daily recharge for each year to generate a 91-year long time series to calculate the annual highest pore-water pressure at each VIC grid cell.  Modeling with maximum recharge provides an indicator of individual storm events that typically trigger shallow landslides (Lu and Godt, 2013), although lesser amounts of recharge may also be sufficient to trigger landslides in some locations.

## 4 Results and Discussion

### 4.1. Geomorphic Analysis and Soil Evolution

Understanding the spatial distribution of dominant geomorphic processes can aid the development of landslide hazard maps consistent with geomorphic theory. In this section, we discuss the mapping of dominant processes on the landscape on the slope and area domain,

and explore the proposed soil evolution model to develop modeled soil depth maps.

### 4.1.1. Investigation of Process Domains

Hillslope diffusion, landslide, debris flow, and fluvial transport processes leave unique imprints on landforms, manifested in the slope-contributing area (S-CA) domain as different scaling

relationships (Montgomery and Dietrich, 1992; Tucker and Bras, 1998; Montgomery, 2001; Stock and Dietrich, 2003; Tarolli and Fontana, 2009).  The infinite-slope factor-of-safety model is only applicable to the initiation of landslides. Therefore, hazards associated with debris flow scour and deposition cannot be predicted by this model. We used a S-CA plot and the infinite slope stability theory to: (1) identify process domains and limit the analysis of the landscape to

slopes where there is shallow landslide potential, (2) evaluate observations of debris avalanches to identify landslide source areas, and (3) infer plausible ranges of the infinite slope stability model parameters to corroborate those we compiled from the literature for NOCA (Table 1).



Our geomorphic analysis was based on plotting, in log-log scale, S, (=tan($\theta$), and CA pairs of each DEM grid cell in NOCA, cells within mapped debris avalanches (including depositional areas), and most likely source areas of landslides identified as the single highest elevation grid cell within each mapped debris avalanche (Fig. 5). The general trend in the S-CA relationship is

acquired for all grid cells of NOCA as well as debris avalanche (DA) cells by binning the data with respect to CA and calculating the mean S for each CA bin. The negative linear relation in the log-log plot suggest a power-law scaling in the form of $S \sim CA^{-B}$ where B is the slope of the S-CA relation on the log-log domain, which reflects channel longitudinal profile concavity. Concavity is generally associated with the role of discharge (CA is used as a surrogate in this plot) in

enhancing sediment transport, while the degree of concavity is tightly related to how nonlinear the dominant transport is with respect to S and CA (Roering et al, 1999; Montgomery 2001; Istanbulluoglu 2009). Geomorphic process domains interpreted from the binned S-CA plot portrayed in Fig. 5 include: (1) a hillslope zone where slope-dependent processes such as dry ravel and soil creep dominate, leading to convex slopes, (2) a landsliding zone where pore-

pressure driven slope failures introduce concavity as landslides arise with shallower slopes as recharge CA grows, (3) a debris flow or saturated landslide zone in headwater channels where mass wasting processes are supplemented with higher fluidity and ground saturation leading to S and CA driven high-concentration transport (Iverson et al., 1997), and (4) a fluvial region where stream-dominated erosion and transport processes ensue (Montgomery and Foufoula-

Georgiou, 1993; Tucker and Bras, 1998). Dominant process domains in the S-CA plot are identified by visual inspection of the scaling transitions that mark changes in concavity. It is well documented that debris flows show reduced concavity relative to both channels and pore-pressure driven landslide zones in the S-CA domain (Montgomery and Foufoula-Georgiou, 1993; Tucker and Bras, 1998; Stock and Dietrich, 2003). The highest profile concavity results from

fluvial transport (Fig. 5).



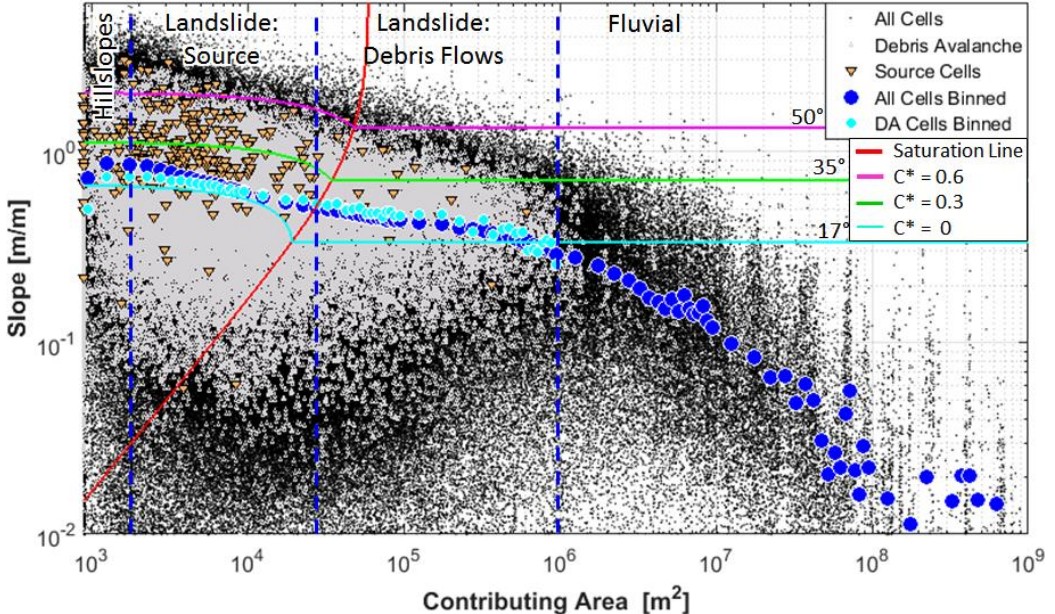

**Figure 5.** Slope-contributing area (S-CA) plot for North Cascades National Park Complex. Mean S for bins of CA are indicated by blue dots and cyan dots for all cells and debris avalanche (DA) cells, respectively. DA source cells (orange triangles) are the single highest elevation grid cell within mapped debris avalanches (gray). Slope stability curves plot the solution of Eq. (1a) for FS=1, given C* and ø=34°. Above each curve landscape is unstable for a given C*. Saturation line (red curve) separates partially saturated areas (left) from saturated areas (right). Vertical lines divide the plot into geomorphic process domains in relation to CA of the landscape (e.g., Montgomery 2001). Cyan horizontal line at 17° generally separates potential landslide dominated areas from fluvial dominated areas.

A threshold CA of approximately 1 km$^2$ and a slope threshold of $\theta$=17° generally separates colluvial mass wasting and debris transport processes from fluvial processes (Fig. 5; see also Legg et al., 2014). Nearly all grid cells within mapped debris avalanches plot to the left of the 1 km$^2$ dashed line. An average $\theta$ value of 17° may also correspond to a low-end of a slope threshold for landsliding. Fully saturated cohesionless soils are unconditionally stable at $\tan(\theta) \leq$ ½ $\tan(\phi)$ (i.e. half of ø), assuming a ratio of water to saturated soil density of 0.5 (e.g., Montgomery and Dietrich, 1994). Solving for ø when $\theta$ = 17° gives 34°, generally consistent with selected ø values from soil texture (Table 1) (Hammond et al., 1992). Approximately 85% of NOCA landscape lies above $\theta$ > 17°, suggesting a dominant role of mass wasting processes in this landscape. We included areas above this slope threshold in our landslide model domain.

The red saturation curve is calculated as $a$R/T, where R/T is calibrated to 0.0005 m$^{-1}$ (e.g., $a/sin\theta$ = 2000 m) to capture most of landslide source cells (left of curve) and a scaling break in the binned S-CA plot (Fig. 5). The saturation curve partitions the landscape into partially saturated (left) and saturated (right) areas, which generally delineates the S-CA pairs separating landsliding from debris flow tracks that form under full soil saturation. For a T = 10 m$^2$ d$^{-1}$, R is 5



mm d$^{-1}$, which is within the range of the lowest maximum annual modeled recharge values in most of the study area, indicating that the plotted saturation line could reasonably map regions that experience saturation annually.

The three lines stacked vertically plot the solution of S in the infinite slope stability equation (Eq. 1a and 2) as a function of CA, and given FS=1, R/T=0.0005, ∅=34° and select values of dimensionless cohesion, C*. Conditioned on the C* value, slopes that plot above the S-CA solution are unstable. Consistent with the binned S-CA data, the solution of the infinite slope stability equation curves down as a function of CA, and following soil saturation, a constant

instability S threshold is reached.   Root cohesion is approximately 6 kPa for C*=0.3 (middle green line) and 12 kPa for C*=0.6 (upper pink line), assuming a soil depth of 1 m.  These root cohesion values are reasonable for shrub and mature forest vegetation found in the literature (Table 1) and they facilitate stability with steeper slopes.  When C*=0 (bottom cyan line), landslides initiate at lower slopes than when cohesion is greater. This solution also envelops the

low slope-end of nearly all landslide source S-CA pairs identified from debris avalanche data. Only a small portion of the unstable areas plot above the C*=0.6 solution of Eq. (1a), which implies areas with higher root cohesion.

### 4.1.2. Evolved Soil Depth

We ran the soil evolution model described in Sect. 2.4 at representative topographic conditions and used the results in a nonlinear regression analysis to estimate soil depth from slope and total curvature.  As the study domain is large, we used a representative population of θ [$^{\circ}$], CA, and Curv values to run the soil evolution model for different vegetation types. The resulting nonlinear equations were used to estimate the mode of modeled soil depth (M-SD) of each

vegetated grid cell of the study domain. Capitalizing on the S-CA analysis (see Sect. 4.1.1), θ, CA, and Curv triplets in each of the CA bins are used from the landscape dominated by colluvial transport processes (θ>17$^{\circ}$ and CA≤1 km$^{2}$).  In order to further classify landscapes within each CA bin, θ and Curv pairs are grouped into shallow (θ ≤ the 10th percentile θ), moderately steep (between 10th and 90th percentiles of θ), and steep (θ ≥ the 90th percentile θ) slope classes.

Within each class, θ and Curv are averaged.

We ran the soil evolution model for 10,000 years to represent the postglacial landscape (i.e., roughly the current interglacial period or Holocene) using the calibrated parameters listed in Table 2, allowing soil sediments to develop from bedrock and to be removed through diffusive

and mass wasting processes. We ran the soil model for the three slope classes and for mature forest, shrubs, and herbaceous root cohesion (Table 1).  Mean and mode soil depth were calculated for a given θ, CA, and Curv for each vegetation type.



**Table 2.** Model parameters used in the soil evolution model

| Parameter | Value | Units |
|---|---|---|
| h(initial) – initial soil depth | 0.01 | m |
| α – rate of exponential decay with depth | 3 | m$^{-1}$ |
| Po – soil production rate from exposed bedrock | 0.0005 | m yr$^{-1}$ |
| Kd – linear hillslope diffusion coefficient | 0.01 | m$^2$ yr$^{-1}$ |
| $\rho_r$ / $\rho_s$ – Rock to soil density density | 2.65/2 | [-] |
| Ks – saturated hydraulic conductivity | 7 | m d$^{-1}$ |
| ø – internal angle of friction | 35 | Degrees |
| Root cohesion[1] | Varies | kPa |
| Recharge (mean)[2] and Coefficient of variation | 32, 0.35 | mm d$^{-1}$ |

[1] Root cohesion varied by vegetation type based on Table 1 and soil cohesion was assumed to be zero.
[2] Recharge extracted from average values found at four representative VIC grid cells within NOCA.

Both θ and Curv have been found to be correlated with soil depth (Heimsath et al., 1997; Braun et al., 2001; Mitchell and Montgomery, 2006; Hren et al., 2007). A multivariate nonlinear regression in the form of $y=\beta_1 \cdot x_1{}^m + \beta_2 \cdot x_2 + C$ was fit to mean and mode of soil depth (predictand, y) given θ and Curv (predictors, $x_1$ and $x_2$) for each vegetation type with R$^2$ >0.9 for all slope classes (not reported). Maps for mode of the modeled soil depth (M-SD) were developed over the portion of the NOCA domain by applying the regression equations using the distributed θ and Curv appropriate for vegetation type at each grid cell. Minimum and maximum depth were set at 0.005 and 2 m, respectively. Outside the colluvial transport process domain are conditions outside the regression analysis; therefore, vegetated areas were assigned a depth of 0.5, 1, and 2 m for herbaceous, shrubland, and forest, respectively, to generate a contiguous soil depth map for NOCA consistent with SSURGO. Areas with barren land cover were assigned a soil depth of 0.05 m, representing the minimum range of modeled herbaceous areas. Developed areas were assigned a value of 0.5 m. Areas assigned a such fixed value are about 2% of the model domain.

As an alternative to the SSURGO-SD, the map of the mode values of M-SD was used to represent the most likely soil depth at each grid cell in the landslide probability model. The evolved soil depth was also used to revise *T*, using the *Ks* provided by SSURGO, which provides a more-distributed continuous field of *T*. The revised T map is used when Landlab is run based on mode from M-SD.

Local erosion is calculated within the soil evolution model. Calibration of the soil evolution model was performed by adjusting model parameters from the literature (e.g., Tucker and Slingerland, 1997; Nicótina et al., 2011) and comparing the mean annual rock erosion rate estimated by the model to long-term average rock erosion rates published for the Cascade Mountains, which range from 0.02 to 0.5 mm y$^{-1}$ over roughly the last several Ma (Reiners et al., 2002, 2003) and slightly higher rates over the last millennia of 0.08 to 0.57 mm y$^{-1}$ (Moon et



al., 2011). In addition to published erosion rates, the resulting soil depths were compared to the SSURGO-SD, which ranged from 0.09 to 2.01 m across NOCA.

Modeled erosion rates ranged from 0.037 to 0.49 mm y$^{-1}$, consistent with published rates as
determined by mineral cooling ages (Reiners et al., 2002, 2003; Moon et al., 2011). In Fig. 6 we show modeled mean annual erosion rates in relation to modeled mode soil depth for a steep and moderate slope class, and illustrate the local variability of M-SD under forest and shrub conditions. The relative frequency histogram of soil depth resembles a triangular distribution, with mode values generally higher than mean values, indicating a negatively (left) skewed
distribution for soil depth (Fig. 6a, 6c). Therefore, there is a higher frequency of deeper soil relative to shallower soils for a given soil distribution. Soil creep fills hollows, thickening soils, as FS gradually drops, leading to episodic landslides that evacuate sediment (Fig. 6b, 6d).

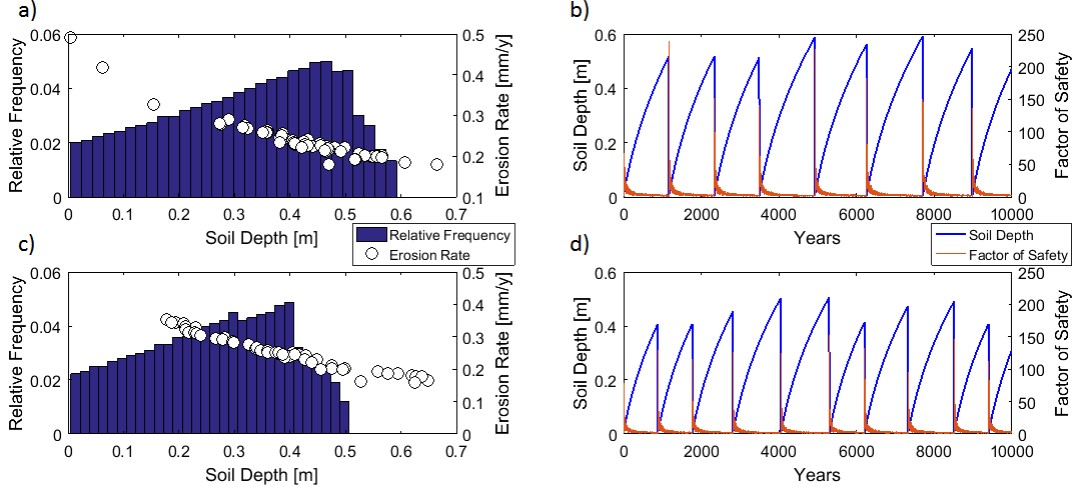

**Figure 6.** Illustration of the soil evolution model run using **(a, b)** steep slope class and forest vegetation
and **(c, d)** moderately steep slope class and shrub vegetation. (a, c) Modeled mean annual erosion rates plotted with respect to soil depth, along with soil depth histogram for a representative convergent location. (b, d) Temporal evolution of soil depth and FS for a representative convergent location with: (a) S=40$^{o}$ and Curv=-0.01; and (b) S=29$^{o}$ and Curv=-0.01.

Comparison of the SSURGO-SD with the M-SD indicates that there is value in a long-term
geomorphic perspective in supplying a spatio-temporal soil depth. M-SD exhibits substantially more spatial variability than the SSURGO-SD (Fig. 7). While both soil depth distributions have similar median values, M-SD has a wider distribution with a higher proportion of shallower and deeper soils than SSURGO-SD. In general, the M-SD is shallower than SSURGO-SD on steeper, convex hillslopes with herbaceous or shrub vegetation and deeper on gentler, concave hillslope
with forest vegetation. For both models, soil depth is greater in the valleys and shallower near the ridge tops (Fig. 7c, d), consistent with other reporting (Anagnostopoulos et al., 2015; Montgomery and Dietrich, 1994).



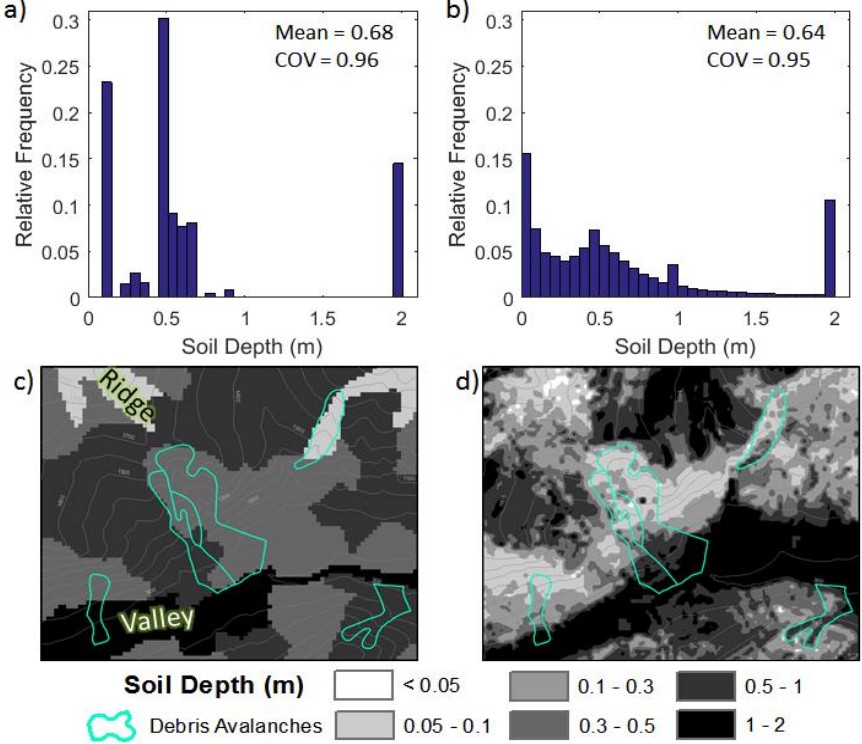

**Figure 7.** Relative histograms of soil depths within NOCA: **(a)** SSURGO-SD and **(b)** mode of M-SD, with respective spatial mean and coefficient of variation (COV). Example location (~6 km$^2$) within NOCA: **(c)** SSURGO-SD and **(d)** M-SD. Mapped debris avalanches are outlined in cyan and contours are at 100-m.

The maximum and minimum soil depth parameters of the triangular distribution to characterize soil depth variability were obtained by analyzing soil evolution model results. At most θ, CA, and Curv triplets using in the soil evolution model, a landslide occurred at least once. Given the negatively-skewed nature of the temporally evolved soil depth, maximum evolved soil depth

was set equal to 10% of the mode in all model simulations. Two M-SD scenarios were developed to compare with SSURGO-SD reflecting existing contemporary and long-term soil depths. In SSURGO-SD and M-SD simulations we set the minimum parameter as 70% of the mode. However, for a long-term evolved soil depth (M-SD LT), if the minimum was greater than 0.005 m, the minimum soil depth was set to 0.005 m, reflecting the effect of landslides

over a long term. This introduces a temporal uncertainty component to modeling landslide probability, which can be used to more accurately estimate landslide return period.

## 4.2 Probability of Failure

Modeled annual probability of failure of shallow landslides, P(F), for NOCA simulated by the Landlab LandslideProbability component using SSURGO-SD and two M-SD scenarios are shown

in Fig. 8. In each run 3,000 values were sampled for model parameters at each grid cell in the Monte Carlo simulations.







**Figure 8.** Landslide annual P(F) map for NOCA overlain with mapped debris avalanches for simulations with: **(a)** SSURGO-SD; **(b)** M-SD; **(c)** M-SD LT. Zoomed-in areas are shown for greater detail in the lower panel in the same order and according to number designated. Purple areas are considered chronically unstable and areas excluded from analysis are shown as gray. Contours are at 100 m. Aerial images of
zoomed-in areas are provided in Fig. 3.

P(F) derived from simulations exhibit low probabilities where slopes are moderate and cohesion is high (e.g., forest). Highly unstable areas largely correspond to steep barren landscape mostly located below retreating alpine glaciers, with steep glacial landforms, transitioning from glacier
to colluvial processes (similar to Guthrie and Brown 2008; Tarolli et al., 2008; Legg et al., 2014) (Fig. 9). Barren areas cover ~13% of the modeled domain. These areas with a thin veneer colluvium, except for moraines, appear to be "continuously sliding" (Borga et al., 2002) or "chronically unstable" (Montgomery, 2001), which also impedes the colonization of vegetation (Dietrich et al., 1995; Istanbulluoglu and Bras, 2005). Shallow soils can enhance the probability
of saturation, leading to high pore-water pressure and saturated overland flows with moderate storms (Pelletier and Rasmussen 2009). Mass wasting activity in barren areas were not completely included in our landslide inventory as they exhibit chronic small-scale slides that do not pose major risks or substantial deposition zones.

20       a)                              b

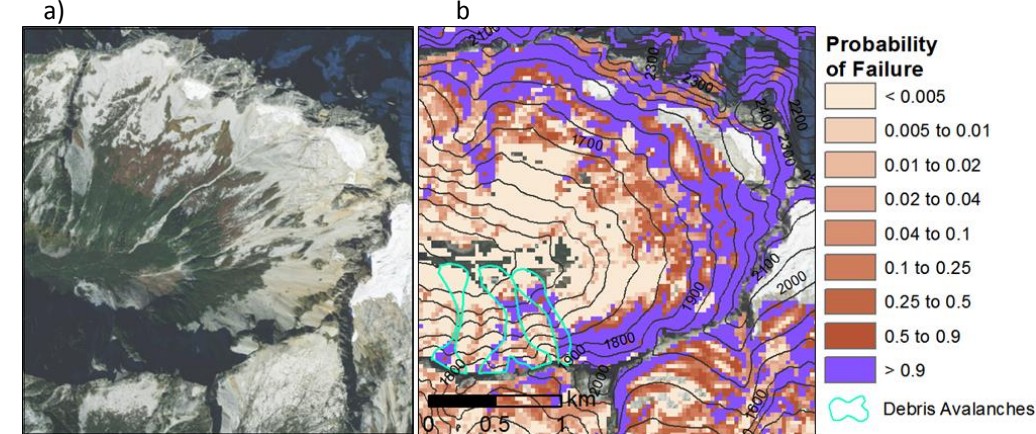

**Figure 9.** Illustration of highly unstable steep areas: **(a)** High resolution (0.3 m) imagery of a NOCA mountain (World Imagery, Esri Inc.)[1] compared to **(b)** P(F) simulated by M-SD with mapped debris avalanches. Contours at 100 m. Notice the barren areas below retreating glaciers with high P(F).

Other locations of higher P(F) are located in topographic hollows (Fig. 8, 9). These converging areas accumulate deeper soils, which decreases the effectiveness of root cohesion, and attract subsurface flow, leading to enhanced pore-pressure (Dietrich et al., 1995). Converging areas often correspond to the upper portions of mapped debris avalanches, which display higher
landslide probabilities than the runout portions in simulations. Thus, the landslide probability visually appears to capture the source area of debris avalanches.



Substantial differences between P(F) derived with different soil depth maps are evident (Fig. 8 and Fig. 10) and corroborate previous studies showing the influence of various soil depth estimates on landslide susceptibility (Dietrich et al., 1995; Okimura, 1998). In general, probabilities are higher and more spatially extensive when the model is parameterized using

SSURGO-SD compared to both M-SD scenarios. Given that other parameters are kept consistent, these differences are attributed to spatial variability of soil depth and related adjustments to transmissivity.

To investigate the spatial distribution of P(F) in relation to soil depth, we plot the cumulative

distribution of P(F), referred to as the fraction of modeled area where P(F) is less than or equal to a given value, for each simulation (Fig 10a). We present our general observations of the spatial distribution of P(F) in the order of SSURGO-SD, M-SD, and M-SD LT as depicted in Fig 8. Simulations show approximately 26%, 38%, and 49% of the modeled domain (79% of NOCA, where θ>17°) as stable (i.e., P(F)=0) under the current vegetation cover and climate. We refer

to these sites as unconditionally stable (i.e., stable even when saturated, and with minimum C and ø sampled) (Pack et al., 1998; Montgomery 2001). A bimodal spatial distribution for P(F) is evident (Fig. 10a, 10b). Areas with low probabilities, around P(F)≤0.1, dominate the spatial distribution of P(F), manifested with a steep rise in the fraction of area from P(F)=0 to P(F)=0.1 (Fig 10a). For P(F)≤0.1 (RP≥10 years), the order of aerial cover for the model domain, including

the stable regions, is 72%, 85%, and 87%. When the unconditionally stable areas are excluded, the percentages become 46%, 47% and 38%, respectively, for the three soil depth products used. This region approximately marks the first peak of the relative histogram of P(F) (Fig. 10b). In the broad 0.9>P(F)≥0.1 range, the increase in fraction of area with P(F) is gradual especially for the two M-SD simulations (Fig. 10a). In the highly unstable regions, with P(F)≥0.9 (RP≤1.1) as

mapped in Fig. 8 and 9, the fractional area begins to rise again in all simulations (Fig 10a). P(F)=1 occupies 11% and 7% of the modeled area in the SSURGO-SD and M-SD simulations, which can be conceptually named as unconditionally unstable (i.e., unstable even when dry and with the highest combinations of C and ø sampled) (Pack et al., 1998; Montgomery 2001). The model run using M-SD LT soil scenario shows a smaller area percentage, ~6%, with P(F)≥0.9,

while SSURGO-SD and M-SD had 16% and 10%. M-SD LT soil scenario provides a more realistic estimate as some locations are not likely to produce slope failures annually due to limited soil development. The second peak of the relative frequency histogram of P(F) appears when P(F)>0.9, largely associated with postglacial barren lands with steep mountain slopes, and converging topography, especially in the case of SSURGO-SD (Fig. 10b). Dominant factors that

control the relative frequency of P(F) are labeled in Fig 10b, and further discussed in subsequent sections.



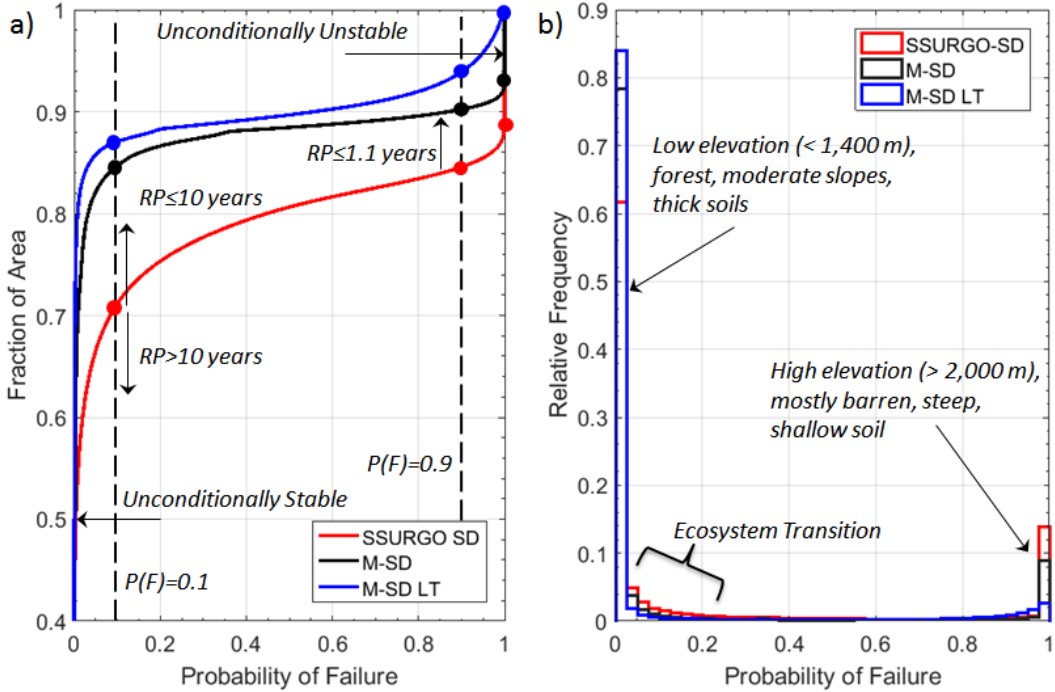

**Figure 10. (a)** Cumulative distribution and **(b)** relative frequency of P(F) (bin size ΔP(F)=0.025) mapped over NOCA from Landlab simulations using SSURGO-SD and two M-SD scenarios. Labels indicate dominant controls on the distribution of P(F) in (b). Fraction of area is used for cumulative spatial probability, plotted using the Weibull plotting position. Return Period for landslides are illustrated only for SSURGO-SD.

We expressed the annual probability of landsliding in the form of a RP, plotted with respect to fraction of area for all three simulations, and mapped RPs for the M-SD LT scenario in Fig. 11. The M-SD LT reduces the probability and increases the return period estimates of landslide initiation, revealing the influence of long-term memory of landsliding on the probability distribution of soil thickness obtained from the soil evolution model. Therefore, the M-SD LT scenario would better suit the definition of RP, while the other two simulations provide reference for relative comparisons. In general and in concert with the P(F), landslides at nearly all RPs affect a greater proportion of the domain when SSURGO-SD is used. Approximately 28% of the model domain is simulated to have a landslide return period of less than or equal to 10 years (i.e., P(F)≥0.1 or frequent slides) based on SSURGO-SD, compared to half as much area, 15%, for simulations using M-SD; M-SD LT had slightly less at 13%. Low return periods (i.e., < 10 years) coincide with steep slopes in barren areas that show chronic landsliding, low-cohesion vegetation type, such as herbaceous, as well as some steep hollows.

At the high end of the return period, 46% of the model domain was simulated to have landslides with a return period of ≥500 years for SSURGO-SD scenario, including stable areas,



compared to 52% and 70% for model runs that used M-SD and M-SD LT scenario, respectively (Fig. 11).   High return periods (i.e., RP>500 years, P(F)< 0.002) are found where slopes are gentler, on divergent topography, and in forest areas.  The fraction of the model domain with a landslide return period between 100 and 500 years is 10%, 18%, and 21% for SSURGO-SD, M-SD, and M-SD LT, respectively, showing a larger fraction in the M-SD products. These landslide frequency rates relate to long-term averages and the actual failures are likely to be clustered in space and time depending on triggering event and the time since the last landslide at the same location (Guthrie and Evans, 2004).

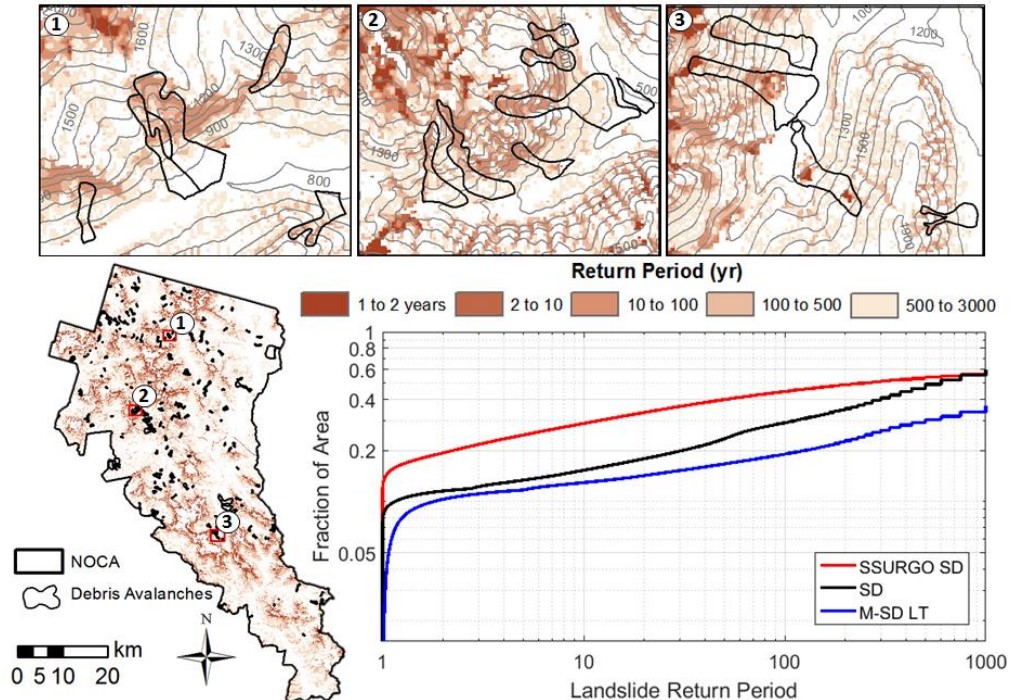

**Figure 11.**  Modeled landslide return period simulations with M-SD LT for NOCA overlain with mapped debris avalanches, including zoomed in areas at top for greater detail. Cumulative distribution of return periods for SSURGO-SD, M-SD, and M-SD LT scenarios, plotted on a log-log scale using the Weibull plotting position.

As soils in landslide locations are formed by sediment accumulation from surrounding hillsides and weathering of the local bedrock, landslides can be the main source of denudation across landslide-prone regions. The expected values of mean annual denudation rate is approximated by: mean(P(F)*$h_s$)/( $\rho_r$ /$\rho_s$) for each simulation. This gives spatial average of the long-term denudation rates due to landslides as 51.9 mm y$^{-1}$, 7.06 mm y$^{-1}$, and 5.04 mm y$^{-1}$ for SSURGO-SD, M-SD, and M-SD-LT scenarios, respectively. While these rates are higher than the reported mean annual denudation rates in this region over the last millennia of 0.08 to 0.57 mm y$^{-1}$ (Moon et al., 2011), M-SD-LT clearly gives the closest estimates to observations among the



three soil depth scenarios. Over an order of magnitude variation in denudation rates is also common as part of long-term records of erosion rates (e.g., Molnar, 2004).

A critical question that remains is: what are the dominant controls that lead to the bimodal
distribution of landslide probability in the modeled domain? First, we examined if topography alone, represented by S and CA pairs, can explain this behavior. The S-CA data pairs from each model grid cell are colored by the value of P(F) in the order from low to high value using output from the M-SD LT scenario (Fig. 12).  As slopes get steeper (S>0.45 or 24.2$^\circ$), a relatively rapid increase in P(F) in relation to slope from 0.4 to 1.0 can be seen, surrounded with lower
probabilities. CA does not seem to impose a visually detectable increase in P(F), which is likely largely due to the wet climate in region. The landslide source cells identified from the highest elevation of debris avalanche shapefiles fall in the "eye" of this high-P(F) region in the S-CA domain. Interestingly, P(F) diminishes in the steepest slopes of most CAs. While the trend of increasing P(F) as slope gets steeper generally shows the influence of slope in Eq. (1a),
landscape with P(F)≥0.4 only constitute about 11% of the model domain (Fig. 10a). For comparison P(F)≥0.1 was 13%. On the other hand, about 57% of the domain has steeper slopes than 24.2$^\circ$ (S=0.45m/m). Locations with slopes less than this are rarely found with P(F)>0.4. This suggest that the majority of the domain with similar pairs of S and CA exhibit lower landslide probability, which can be largely attributed to the spatial distribution and influence of
vegetation type and soil depth (e.g., Roering et al., 2003).

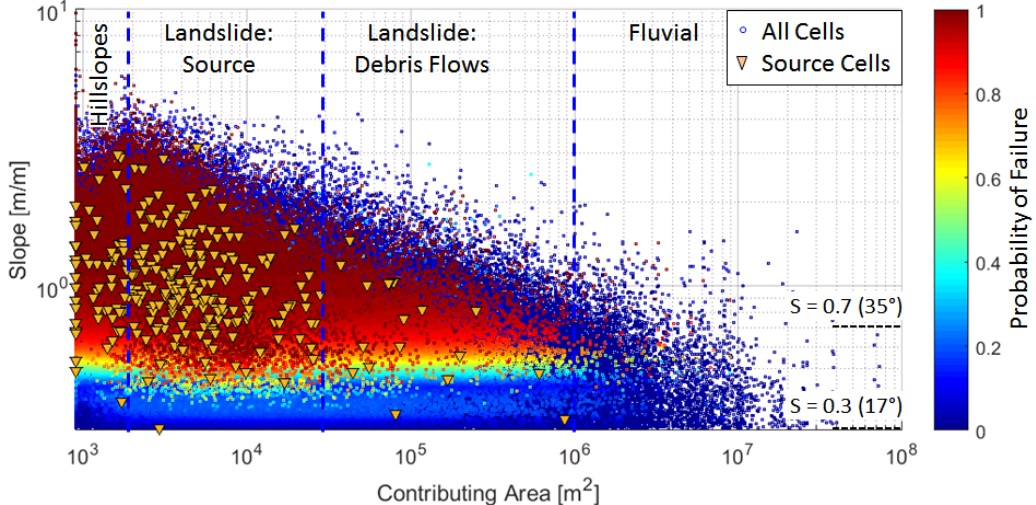

**Figure 12.** S-CA plot colored by the P(F) simulated with from the M-SD LT.  Source cells (orange triangles) are the single highest-elevation grid cell within mapped debris avalanches.  Comparable to Fig. 5. High
probabilities plot over low probabilities.

We investigated the roles of vegetation, slope steepness, and soil depth on P(F) in relation to elevation (Fig 13). From low to high elevations, vegetation changes from predominantly forest (elevation <1,400 m) to coexisting shrub, herbaceous plants, and barren land (1,400 m to 2,200





m) as a result of elevation-dependent ecoclimatic controls (e.g., temperature) on vegetation survival and growth (Fig. 13a). In this region of ecosystem transition, the mean P(F) shows a persistent increase from 1,400 m until a maximum is reached between 2,200 and 2,400 m, depending on simulation (Fig. 13b, 13c). Observations of debris avalanche by elevation confirm

the pattern of P(F) dependence on elevation in relation to ecosystem change; 75% of the extracted landslide initiation zones from mapped debris avalanches are located between 1,200 m to 2,000 m (Fig. 3b). In the 1,400 to 1,900 m elevation range of the ecosystem transition zone, mean slope is relatively constant ~0.75 m/m (~37°), and rises up to 0.9 m/m (42°) between 1,900 and 2,200 m (Fig 13c), consistent with the binned-averaged slopes of the

landslide source area in the S-CA plot in Fig 5. Mean soil depth begins to drop in both SSURGO and modeled soil depth products above 2,200 m.

These observations confirm the strong control of ecosystem transition on landslide activity in the region. Below about 1,400 m (~40% of NOCA), forested vegetation combined with deeper

soils and moderate slopes keep P(F) low. In the 1,400 to 2,200 m range, loss of root cohesion with ecosystem transition combined with gradual increase in landscape slopes contribute to increased P(F). Above 2,200 m elevation, soils become very shallow and slopes exhibit the steepest angles in the modeled domain. This combination leads to the largest variability in P(F), combining the highest P(F) values, P(F)≥0.9 mostly attributed to barren areas (~6% of the model

domain in the M-SD LT scenario), with lower P(F) values where thinner soils reduce the driving force within Eq. (1a). In aggregate, thinner soils at higher elevations lead to lower mean P(F), which we referred to as soil depth control. The general contribution of elevation on the spatial organization of P(F) is labeled in Fig 10b.



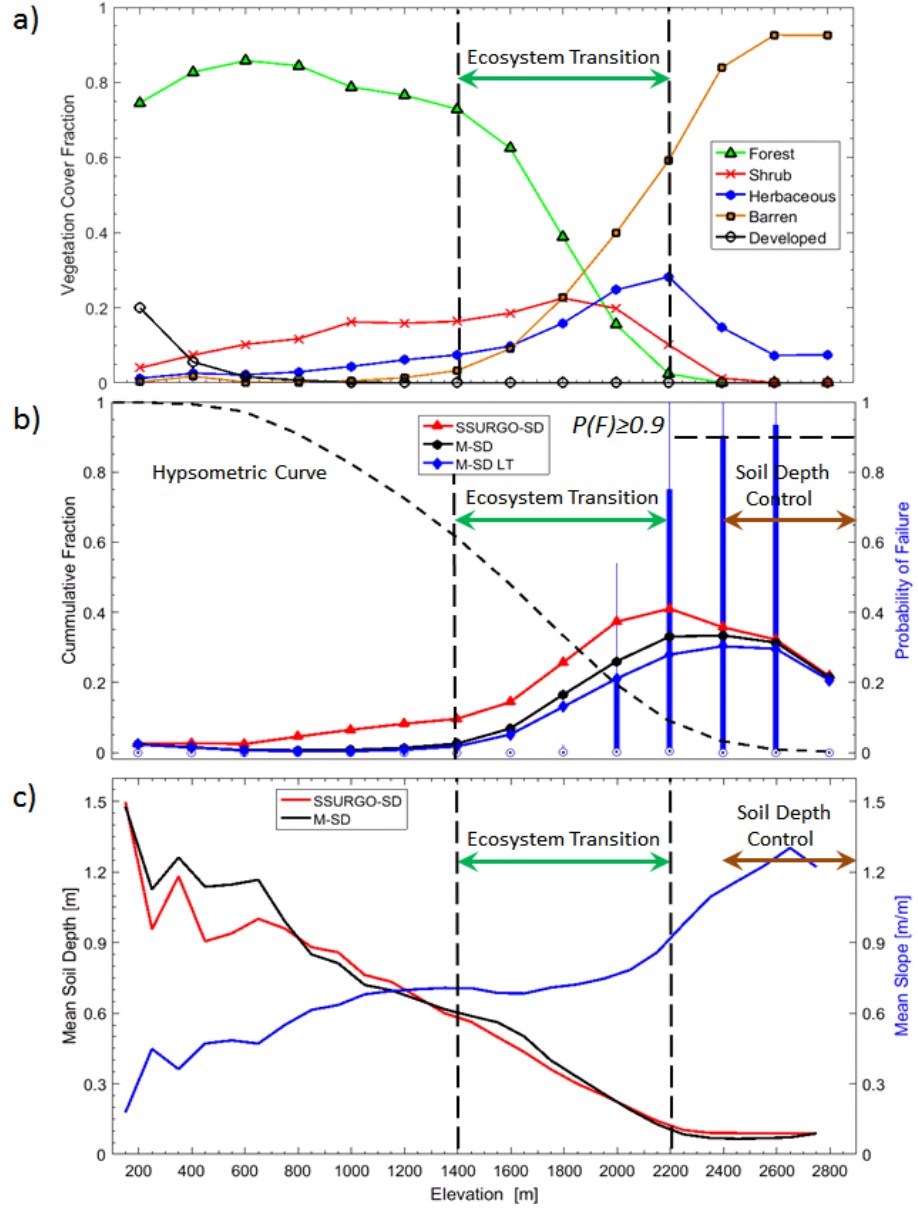

**Figure 13.** Elevation (200 m bands or bin) influence on: **(a)** vegetation cover fraction for NOCA, taken as fraction of vegetation type within each elevation band, **(b)** mean P(F) using SSURGO-SD and two M-SD scenarios, along with compact box-whisker plots for P(F) of M-SD LT scenario where circles-dot symbol represents median (outliers not shown, greater than 1.5 interquartile distance) overlaid with hypsometric curve for NOCA, and **(c)** mean soil depth for SSURGO-SD and M-SD products with mean slope. Mean values calculated within each 200-m elevation band.



### 4.3 Model Evaluation

The performance of a landslide model is often based on its ability to capture existing mapped landslides.  In Sect. 4.2 we evaluated our model through visual comparison of modeled P(F) to observed landslide locations (e.g., Fig. 8, 9). In this section, a more quantitative approach is presented for model evaluation.  We statistically evaluated our model using multiple approaches, including cumulative distribution (CD) of P(F) comparisons as well as Receiver Operating Characteristics (ROC) (Fawcett, 2006) and Success Rate (SR) curves (Bellugi et al., 2015).

For the statistical analysis, we limited our performance assessment to the source areas of the mapped debris avalanches.  Source areas of debris avalanches were not mapped separately from the remaining debris avalanche features (i.e., transition and deposition zones) hindering the evaluation of model predictions (Tarolli and Tarboton, 2006). Because we anticipate that the source areas could originate from a neighboring grid cell, we considered source cells as a collection of grid cells in the upper 10, 20, and 30% highest elevation cells within each mapped debris avalanche shapefile.  These populations of source cells were treated as 'observed' landslide source cells during validation of the landslide probability using CD, SR and ROC performance metrics. In this validation, we excluded barren areas with slopes ≥ 37° (~5% of the model domain), which characterizes slopes of active small-scale dry landslides (failure depth ≤ soil depth) more appropriately represented by nonlinear hillslope diffusion models (see Roering et al., 1999; DiBiase et al., 2010; Pelletier et al., 2013).

For comparison of P(F) with source area cells, we randomly sampled 50,000 grid cells outside mapped debris avalanches (~2% of the modeled domain), similar to the number of grid cells within entire mapped debris avalanche areas.  The majority of the source grid cells and outside debris avalanches cells are located at elevations between 1200 and 1800 m (Fig. 14a). Grid cells in the random sample outside debris avalanches were constrained to the elevation range of the source cells to allow unbiased comparison. We recognize that the areas outside mapped debris avalanches have the potential to be unmapped landslides, other landslide types, or unstable areas deficient a triggering event; therefore, we interpret the test results conservatively.  We expect the simulated P(F) should estimate lower probability outside debris avalanches compared to source areas of mapped debris avalanches.

At low and mid elevations, simulations generally showed a greater fraction of high probabilities in source areas compared to outside of debris avalanches (Fig. 14b, c).  However, when only high elevation (>1,800 m) data were considered, the pattern was reversed with a larger fraction of high probabilities found outside debris avalanches than source areas (Fig. 14d).  At this higher elevation, much of the land cover is barren or herbaceous (i.e., low root cohesion), resulting in high probabilities of failure throughout the model domain (Fig. 12a). While there are extensive shallow failures in these regions only limited amount of those that turned into debris avalanches were mapped.  This reverse pattern is also present at mid-elevations (~1,200 to 1,800) for both M-SD scenarios, only for 10% of the sample data, when P(F)>0.03 and





P(F)>0.1 for M-SD and M-SD LT, respectively (note the crossing curves in Fig. 14b). At low elevation (~125 to 1,200 m), there were no source areas with P(F)>0.4 in M-SD and M-SD LT scenarios.

The performance of the model results we are presenting in this paper are specific to a sample comparison of 10% source area of mapped debris avalanches and random sampling outside debris avalanches. When examining the validation datasets in their entirety (i.e., regardless of elevation), the median P(F) of the 10% source DA cells is 13 times the median P(F) of outside DA for SSURGO-SD and four times the median P(F) for M-SD; median is zero for outside DA cells in
the M-SD LT. The Kolmogorov-Smirnov test (Chakravart et al., 1967) test show paired comparison between DA source area cells and cells outside DA for all three scenarios are statistically different (p<<0.01).

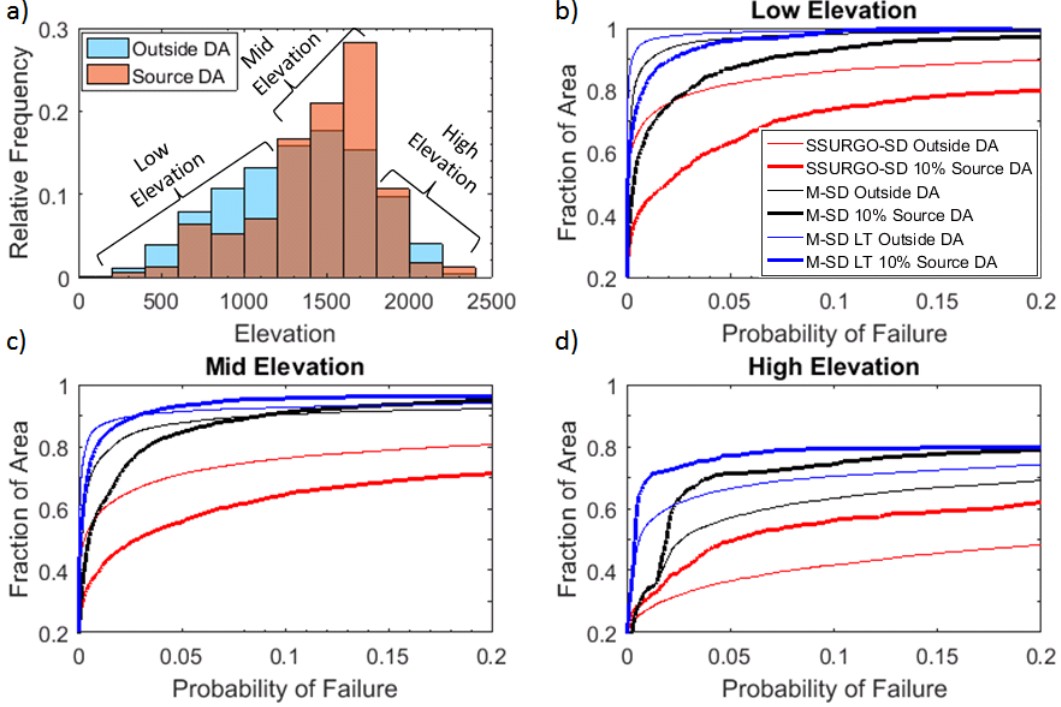

**Figure 14.** a) Relative histogram of source areas in upper 10% elevation of debris avalanches (DAs) and
for 50,000 grid cells outside DAs. Cumulative distributions of P(F) plots limited to P(F) ≤ 0.2, or return period ≥ 5 years, to highlight detail in simulation using SSURGO-SD, M-SD, and M-SD LT at: b) low (<1,200 m), c) mid (1,200 to 1,800 m), and d) high (>1,800 m) elevations as depicted in a). Thicker lines represent probabilities for source areas of (DAs) and thin lines represent probabilities for cells outside DAs.

Another statistical analysis uses ROC curves to examine how our model compares with randomly distributed landslides over the landscape. These curves are constructed from confusion matrices generated from comparisons between observed and modeled landslides,





based on varying P(F) threshold (e.g., 0.1, 0.2, 0.3, etc.). True positives (TP) are those cases within observed landslides where probabilities are equal to or greater than threshold. False negatives (FN) are probabilities within landslides that fall below the threshold. False positives (FP) occur outside observed landslides with simulated probabilities equal to or above the

threshold. True negatives (TN) are also outside observed landslides, but with probabilities below the threshold. From these metrics, true positive rate (TPR) or fraction of landslides captured and false positive rate (FPR) or fraction of false alarms can be calculated as follows:

$$TRP = \frac{TP}{(TP+FN)} \tag{8}$$

$$FRP = \frac{FP}{(FP+TN)} \tag{9}$$

The advantage of the ROC curve over a standard confusion matrix is the ability to vary the probability threshold for assigning model simulations to a modeled landslide (positive) or no landslide (negative) classification, generating different positive and negative comparisons (Mancini et al., 2010; El-Ramly et al., 2002; Anagnostopoulos et al., 2015). A better performing model will exhibit a curve toward the upper left of a FPR (x-axis) and TPR (y-axis) plot. A 1:1 line

in the plot represents a trivial model that randomly assigns stable and unstable cells. The area under the curve (AUC) generated by ROC curve quantifies the performance of a model for identifying landslide and non-landslide locations. The AUC statistic represents the probability of correctly ranking a landslide and non-landslide pair randomly selected from those two datasets (Hanley and McNeil, 1982). SR curves are similar to ROC curves, with TPR as the y-axis,

but compares this to the fraction of landscape predicted as unstable (x-axis), calculated as (TP+FP)/(TP+FP+TN+FN). Again, a relatively well performing model would plots farther away from the 1:1 line representing a trivial model.

For this comparison, we used the same datasets used in the cumulative probability analysis
discussed Sect. 4.2. Both simulations using SSURGO and M-SD modeled 10% source areas and non-landslide areas better than random selection as demonstrated by the curves plotted above the 1:1 line (Fig. 15). The classification is stronger as the source area fraction is reduced. However, the model's strength in the classification is modest as indicated by the AUC values of between 0.60 and 0.61, compared to an AUC of 1 representing a perfect classification. The

TRIGRS-P probabilistic landslide model tested by Raia et al. (2014) found higher AUC results (i.e., 0.65 to 0.73). However, their study tested small areas (3 to 6 km$^2$) that were well studied locations with detailed inventories of landslides resulting from one or two winter rainfall seasons and the entire landslide was tested rather than source areas only.





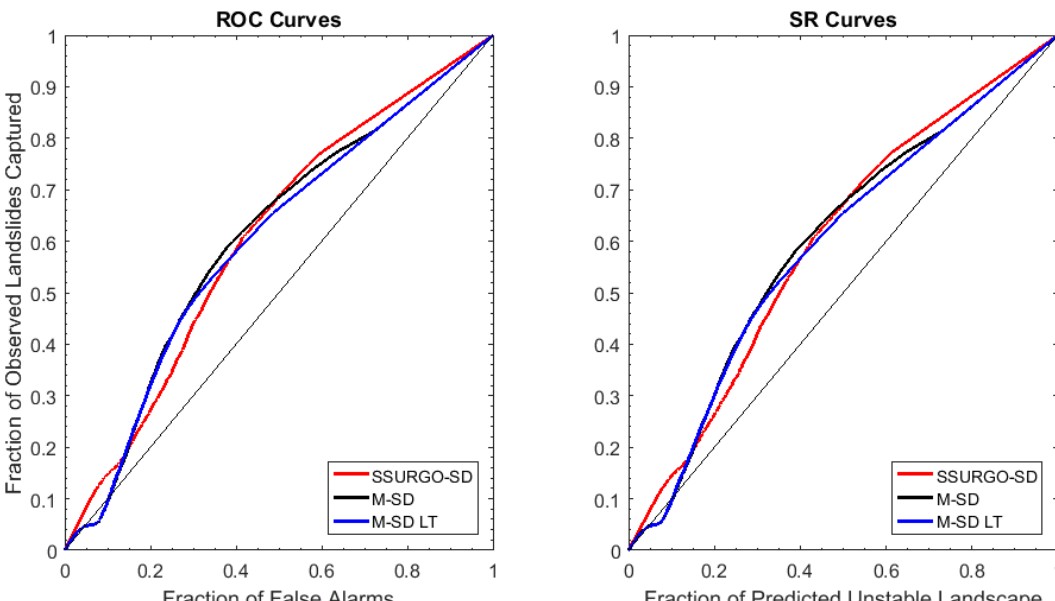

**Figure 15.** a) ROC curves and b) SR curves for simulations using SSURGO-SD, M-SD, and M-SD long-term (LT). Comparison represent P(F) for the upper 10% of DA as observed landslides to a random sample of 5,000 cells outside DAs. Thresholds for simulated probabilities associated with positive classification of a source areas declines along the curves from lower left to upper right. Black diagonal line on a 1:1 line represents the case of a trivial or random classification model.

ROC and SR curves provide an indication of how well the modeled simulations of P(F) classify both observed landslide source cells and non-landslide grid cells compared to random classification. The crossing of ROC and SR curves in the simulations with M-SD (Fig. 15) implies that at higher probability thresholds, simulated probabilities delineate more false alarms (e.g., areas outside DAs as unstable) than capturing source areas. This may be indicative of the high probability values at high elevations even outside the debris avalanches where vegetation is sparse, as was indicated above in the analysis of cumulative distribution plots. We found for our case study that the optimal probability threshold to maximizing landslides captured and minimizing false alarms (i.e., point around the apex of the ROC curves) declines by half depending on the simulation: P(F)≥0.008 (i.e., RP≤125 years) for SSURGO-SD, P(F)≥0.004 (i.e., RP≤250 years) for M-SD, and P(F)≥0.002 (i.e., RP≤500 years) for M-SD LT.

The modeled potentially unstable landscape has generally been greater than observed landslides when infinite slope stability models are calibrated with limited observations (Sidle and Ochiai, 2006; Baum et al., 2010). As pointed out by Borga et al. (2002), concluding "overrepresentation" of areas potentially subject to shallow landsliding can be misleading because the absence of mapped landslides does not necessarily indicate an absence of landslide hazard over time across the landscape. Locations with high landslide probability




outside mapped landslides in both simulations could be indicators of where to conduct additional investigations for missed landslides or areas on the verge of failing.

Validating hazard maps is challenging, especially in large areas of remote mountainous regions, because inventories are typically incomplete, lack the date of landslide occurrence, different landslide types likely have different meteorological triggers, environmental conditions change after a landslide event, and unidentified high probability areas may fail in the near future even though they appear to be stable during an inventory (van Westen et al., 2006;Tarolli and Tarboton, 2006).  Additional evaluation of model performance would benefit from field
investigation in areas of high and low modeled P(F) to identify any landslides or instability that may have been missed during the original inventory.  Future work that couples the volume of sediment available for landsliding will lead to further improvements in estimating hazards and potential impacts from landslides.

## 4.4. Model Limitations
For model design and computation efficiency, we made several simplifying assumptions.  We neglect groundwater leakage to the bedrock in recharge estimation and apparent soil cohesion through the effect of surface tension in unsaturated zones, both of which could be added to future updates to the component.  Tree and snow surcharge is also disregarded, although it
may have some stabilizing effect where soils are shallower than 1 m (Hammond et al. 1992).  Our approach does not simulate the actual number of landslides, landslide type, nor the size of the landslide because the discretized nature of the failure field precludes specific knowledge of which and how many grid units may be involved in a failure at a particular time.  These model omissions present opportunities for future customization of the component or coupling with
other models.

Modeled probability does not capture the runout of debris avalanches, which can travel considerable distances in steep mountainous environments.  Some unexpected results depicted higher probability in runout portions of some debris avalanches when using SSURGO-SD, but
these probabilities were lower when M-SD scenarios were used (e.g., Fig. 8, middle zoomed-in panels).  Mis-mapping of probabilities of failure and observed landslide are likely attributed to variations in soil depth, material properties, and hydrologic routing (Schmidt et al., 2001).  Model parameters such as slope derived from DEMs developed with post-landslide mapping can also contribute to reduced probabilities in observed landslides where slope and soil depth
were reduced.  Furthermore, inventories over broad areas are challenging as landslides are isolated processes that may occur with regularity, but may not be large in size (Van Westen et al., 2006). Finally, steady-state flow that we used for subsurface flow neglects transient processes and roles of macro-pores. Macropores from decayed roots or animal activity can be important in transporting water relatively quickly from the surface to deeper soil layers and
groundwater (Sidle et at., 2001; Gabet et al., 2003; Beven and Germann 2013).



## 5 Conclusion

We develop a regional model of probabilistic shallow landslide initiation based on the infinite slope stability equation coupled by steady-state subsurface hydrology driven by groundwater recharge. Uncertainty in model parameters is explicitly accounted for through Monte Carlo
simulation. A geomorphic soil evolution model provides a spatially-distributed soil depth alternative to homogeneous patches of soil depths provided by SSURGO. This feature allows the landslide model to be used where soil depth information is uncertain, sparse, or absent. Our model developed in Landlab (Hobley et al., 2017) is made up of a landsliding component, a Landlab utility for hydrologic data processing, and a model driver that runs the component.
The model driver can be run on personal computers or online via Hydroshare through cloud computing creating reproducible results. Our approach demonstrates:

- Regional maps of landslide hazard produced with three different soil depth scenarios reveal alternative simulations of probability of landslide initiation, reflecting the
importance in soil depth in landslide hazard prediction.
- Simulations using SSURGO-SD returned higher probability of failures and shorter return periods than simulations using modeled soil depth products (M-SD and M-SD LT). The M-SD LT simulation further reduces the probability of failure and increases the return period. Mean annual denudation estimates from the M-SD LT scenario show closer
estimates to published rates of denudation over the last millennia than the other simulations.
- SSURGO-SD scenario provide a short-term tool for high risk planning using conservative estimates of probability of failure, while M-SD LT provides long-term estimates more consistent with landslide frequency in the region and useful for management of
ecosystems and aquatic habitats, and estimation of sediment budgets for watershed planning.
- Elevation dependent patterns in probability of landslide initiation show the stabilizing effects of forests in low elevations, an increased landslide probability with forest decline at mid elevations (1,400 to 2,400 m), and soil limitation and steep topographic controls
at high alpine elevations and post-glacial landscapes. These dominant controls manifest in a bimodal distribution of spatial annual landslide probability, modes controlled by highly stable forested and chronically unstable post-glacial domains and other barren areas.
- Model testing with limited observations revealed similar model confidence for the three
hazard maps, suggesting suitable use as relative hazard products. Validation of the model with observed landslides is hindered by the completeness and accuracy of the inventory, estimation of source areas, and unmapped landslides.
- Our shallow landslide hazard model provides regional scale estimates of the relative annual probability of shallow landslide initiation as well as landslide return period,
which is useful for civil protection through land use planning to minimize geohazard consequences from precipitation triggers.



## 6 Data and Model Availability

To facilitate ease of use of the landslide hazard model, we developed the landslide model as a component of Landlab, an open-source Python toolkit for two-dimensional numerical modeling of Earth-surface dynamics available at GitHub: http://github.com/landlab/landlab (Hobley et
al., 2017). Documentation, installation instructions, and software dependencies for the entire Landlab project can be found at: http://landlab.github.io/. The Landlab project is tested on recent-generation Mac, Linux and Windows platforms using Python versions 2.7, 3.4, and 3.5. The Landlab modeling framework is distributed under a MIT open-source license. A user manual and driver scripts for the application of the Landlab LandsideProbability can be found
at: https://github.com/landlab/pub_strauch_etal_esurf (Strauch, GitHub Repository)

Online access to the Landlab LandslideProbability model is freely provided through https://www.hydroshare.org, where data and code drivers are available to demonstrate and explore the model using interactive IPython notebooks in a JupyterHub. Thus, users can access,
test, adapt, and apply the landslide model for their area of interest without downloading Landlab or the components. Data and driver code used in this analysis are available at hydroshare (Strauch et al., 2017). Existing demonstration driver codes can be adapted to fit data provided in raster format by the user to create distributed data fields used as parameters in the component. Instructions for accessing HydroShare and the online demonstrations, codes,
and data used in this paper are provided in supplemental material.

### Acknowledgements

This research was supported by the US National Science Foundation (CBET-1336725, OAC-1450412) and USGS Northwest Climate Science Center. We thank Dan Miller for helpful review
of an earlier version of the manuscript. Technical editing on portions of the manuscript was provided by Brad Strauch and Diann Strom. We also appreciate the developers of Landlab, including open-source contributors to the earth surface processes modeling community. Data repository (Strauch et al., 2017), testing, and online reproducibility was facilitated by the cyber infrastructure of HydroShare services provided by researchers associated with the Consortium
of Universities for the Advancement of Hydrologic Science, Inc. (CUAHSI), particularly the support of Tony Castronova, and the team at CyberGIS Center for Advanced Digital and Spatial Studies for their maintenance and support for our use of the ROGER Supercomputer at the National Center for Supercomputing Applications (NCSA) at University of Illinois at Urbana-Champaign.

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
