# Peer review of "A hydro-climatological approach to predicting regional landslide probability using Landlab"

_Earth Surface Dynamics, 2017_

## Author Comment (AC1) · 22 Jun 2017

In Sect 6 Data and Model Availability (line 10, pg 40), the GitHub link to the user manual and driver scripts is incorrect. The correct link is: https://github.com/RondaStrauch/pub_strauch_etal_esurf. Sorry for the inconvenience.

---

## Referee Comment (RC1) · Anonymous Referee #1 · 1 Aug 2017

The paper describes a integration between a physically-based hydrological model based on a steady-state subsurface flow representation and the infinite slope stability model. The model, which is available as a component in Landlab, an open-source, Python-based landscape earth systems modeling environment, has been applied in a 2,700 sq km steep mountainous region in northern Washington, (USA) using 30-m DEM resolution. The model structure is not a novelty (i.e. it is similar to other models like the SINMAP) even if the authors try to use a hydrological model (VIC) in order to derive the recharge R. The authors derived three maps probability of landslide initiation highlighting the presence of elevation dependent patterns. A model validation has been carried out using observed landslides showing performances, which have been declared as "modest" by the same authors. I find the research scope generally

good. The concepts are almost always sufficiently exposed, the topic is important in the hydrology field with a medium-high impact for the hydrological-geomorphological sciences. However, in my opinion the paper could be accepted after a major and careful revision. In the following I report my observations:

Main problems

1. Manuscript structure: the paper is long and a little bit convoluted. In my opinion the length of the paper could be reduced without any important loss of information and the structure could be improved. For example the sections related to the cyberinfrastructure Landlab (sect. 2.2 and figure 3 and the last part of the section 2.3) could be reduced or removed since it is less important for the reader of ESurf (see Aims and Scope of the ESurf Journal). Some parts are difficult to understand (see other comments) and there are some repetitions that can be removed.

2. Basic assumptions:

a. The authors fixed a soil density equal to 2000 kg/cubic meter constant over the entire domain; is this relative to the bulk density of the soil or to the wet soil density? Is this assumption realistic considering that you have different soil type in your domain?

b. The authors assume the soil as incoherent (C=0) assigning all the cohesion to the root? Again is this assumption realistic? Please consider that also a loamy sand could provide a cohesion greater than that given by the root system. Please try to justify this assumption using field data relative to the soil mechanics parameters.

c. The authors assume that the recharge is given by the sum of the baseflow and the surface runoff (page 10) at each VIC grid cell. It is not clear the reason of such an assumption since usually the recharge is given only from the subsurface flow (i.e. part of the baseflow) as highlighted by the authors (page 6 – line 23-25). In similar modelistic approach (SINMAP) R is considered as a climatic factor (rainfall). Please clarify this apparent contradiction.

d. It is not clear how the soil depth evolution model and the stability model are coupled (if they are coupled). I thought that the outcome of soil depth evolution model is provided as soil depth map (in terms of mode) but there is a sentence (page 11 – line 26-27) which is in contrast with my previous thoughts ("Eq (a) and Eq (2) are used to calculate FS within the soil evolution model". So please try to clarify the connection between these models. I think that a figure with a flow chart describing models and connections together with the setup of the experiment could be useful to the readers. How many simulations did they run?

3. Choice of model parameters: the choice of geotechnical and soil parameters (mode and range of variability used for MonteCarlo simulation) is, at least, not convincing.

a. The internal friction angles are fixed in Table 1 in terms of mode, min and max. I'm not convinced by these values; they seems to be very high especially for the loamy sand and sandy loam. Could you provide references or field data used to fix these values?

b. The authors use different relationship to define minimum and maximum value of trasmissivity T and friction angle. How do they define these relationships? What is the impact of these values on the final results (sensitivity analysis).

4. Low performance of hazard maps: the authors affirm that the performance of proposed approach is modest. I agree with them and if the aim of this model is to create a map of landslide hazard, better results could be achieved using classical susceptibility approach based on statistical methods or data-driven methods. Moreover I think that they can remove the CD approach to test the performance of the proposed approach. The CD approach is aimed to highlight the existence of a statistically significant difference between the two P(F) cdfs (within and outside) for fixed soil depth scheme. The authors can only affirm that the two cdfs are different but this does not imply that the model performances are acceptable. I understand that this could be a first level check, but I think that can be removed without any problem for the paper.

Other comments

* Page 7, lines 9-10: The sentence is not clear. How does the use of maximum annual daily recharge help to define uncertainty in R?

* Page 7, equation (3a): Please define n and n(FS<1).

* Page 9, line 9-21: The difference between options 2 (lognormal) and 3 (lognormal spatial) is not enough clear. Also the option 4 is not clear. Please clarify this paragraph.

* Page 9, line 30: what is core node? Is it a computational element of spatial domain? I think that these details on the computational framework are not necessary since they create a little bit confusion in the main line of the paper.

* Page 10, lines 16-19: please check the sentence since it is not very clear.

* Page 15, line 11: define combined curvature (reference). Is it different from the total curvature used further?

* Page 16, line 22-23: sentence not clear; lines 23-25: this sentence can be removed.

* Page 17, lines 7-8: what is the meaning of "spatially consistent". You can use consistent when map is compared with the field data. Did you carry out this task?

* Page 17, lines 18-19: how can the authors "confirm" the ranges of soil depth used through a long term evolution model which needs to be calibrated on soil data as well?

* Page 19, lines 15-17: how the authors calculate pore-water pressure starting from the maximum daily recharge? Do they use pore-water pressure in their stability model? I think they use directly the recharge (see equation 2).

* The section 4.1.2 is not clear especially the role of regression based equation which seems to provide the soil depth as a function of slope and curvature. If you run the soil evolution model, why do you need a regression to obtain soil depth? In the same section it is not clear the difference between M-SD and M-SD LT. I think that there is a

lot of information but this is not well-organized.

* Figures 10 and 11 highlight the same information (Probability or return period). Please consider removing one of the two figures.

* Page 32, line 13: The authors are not using "observations" in figure 13 but model results. Please change the sentence.

* Figure 13c: in the legend line relative to M-SD LT is missing.

* Page 34, line 16: since the authors use 10% of highest elevation cell, I suggest to remove 20% and 30%.

* Page 34, lines 24-26: I suggest specifying the number of DA source cells and the number of DA outside source.

* Figure 14a: I think there is an error in the plot. The sum of all the bars must be equal to 1. If this is true for Outside DA, it cannot be true for Source DA since for each bin the relative frequency is lower.

---

## Referee Comment (RC2) · Anonymous Referee #2 · 8 Aug 2017

Dear Editor, dear Authors,

I've read this article for possible publication in Earth Surf. Dynam. The work describes an integrated python-based tool for the evaluation of regional landslide probability initiation throughout a hydro-climatological approach.

The work is of high scientific value and the applied methodologies are scientifically robust. However, to my opinion, the paper ended up being excessively long, sometimes not immediately clear and the overall application not well focused on clear and simple targets. Moreover, given the numerous details of the developed numerical model system, I wonder if a journal which addresses to models development or environmental software would be more appropriate. Anyways, to my opinion, it needs major revision for it to be published in order to make it clearer, more fluent, to better define the aims

of the application and to improve the literature review which lacks of some important contributions. Please, read in the following my main concerns.

1. The manuscript is very long and sometimes repetitive, with English style a bit verbose. I have to read it a couple of times to get to the point. Some parts can be synthesized and stated more directly.

2. Literature review is well done and comprehensive of various aspects involved in this work. However, it also lacks of some important contributions in the specific field of physically based modeling for rainfall-triggered landslides, also with regard with the parameters uncertainty.

Please read throughout the following comments some suggestions.

3. My main concern is whether simpler and more computationally efficient statistical approaches for susceptibility evaluation could be more appropriate for such regional and long term analysis. The methodology is based on the use of various simplified models which make it complex. The ultimate model performances are not very satisfactory in terms of ROC and AUC. The approach is classified as dynamic and processed-oriented; however it is not able to reproduce and simulate specific events due to the simplifications and the large temporal scale and can be used only for long term analysis, thus becoming a kind of 'static' approach. Statistical models are very robust and able to guarantee very satisfactory results (e.g. Lepore et al., 2012; Lee and Pradhan, 2007).

Lepore C, SA Kamal, P Shanahan, RL Bras (2012). Rainfall-induced landslide susceptibility zonation of Puerto Rico. Environmental Earth Sciences 66 (6), 1667-1681

Lee, S., Pradhan, B. (2007). Landslide hazard mapping at Selangor, Malaysia using frequency ratio and logistic regression models. Landslides 4, 33-41

4. In the model system there is a mix of temporal and spatial resolutions (soil depth evolution at yearly scale, hydrological model at daily and 6 kmq square, geomorphological model at 30 m). If I understood well the finest temporal resolution is the daily scale of the annual maximum recharge. However, the daily resolution misses the most intense events and moreover, the daily annual maximum recharge does not guarantee the worst 'hydrological conditions' since the antecedent soil moisture conditions are also influent. This why I am skeptical on the advantage of this approach instead of others (comment 3).

5. Soil depth evolution: it is not totally clear to me whether the soil depth evolution model is run in conjunction with the stability module or it is run 'off line' and the final map is then fed to the slope stability module. Also, how the soil depth evolution influences the hydrological module? Theoretically, the soil evolution model itself should take into account the change in elevation due to the landslide. Is this done? Please make it clear.

6. Authors do not explicitly discuss the importance of the effect of matric suction and the 'apparent' cohesion which arises under unsaturated soil moisture conditions (e.g., Simoni et al., 2008; Baum et al., 2002) and which can be much higher than soil and also root cohesion (e.g., Arnone et al., 2016). They discuss clearly hypothesis of steady state conditions, but this does not justify the neglecting of the matric suction. Moreover, several procedure have been also proposed to predict shear strength under unsaturated soil (based on modified Mohr-Coulomb failure criterion (Vanapalli et al., 1996; Fredlund et al., 1996)), which have been used in various works (Montrasio and Valentino, 2008; Lepore et al., 2013). I suggest referring to Lepore et al., (2013) for a discussion on this point.

Arnone E, Caracciolo D, Noto LV, Preti F, Bras RL (2016) Modeling the hydrological and mechanical effect of roots on shallow landslides. Water Resour Res 52(11):8590–8612

Baum, R. L., Savage, W. Z., and Godt, J. W. (2008) TRIGR-a Fortran program for transient rainfall infiltration and grid-based regional slope-stability analysis, US Geological Survey Open File Report 2008-1159, 75 pp.

Fredlund, D. G., Xing, A., and Barbour, M. D.(1996). The relationship of the unsaturated soil shear strength to the soil water characteristic curve, Can. Geotech. J., 32, 440–448

Lepore C, Arnone E, Noto LV, Sivandran G, Bras RL. (2013). Physically based modeling of rainfall-triggered landslides: a case study in the Luquillo forest, Puerto Rico. Hydrology and Earth System Sciences 17: 3371–3387. DOI: 10.5194/hess-17-3371-2013.

Montrasio, L. and Valentino, R. (2008) A model for triggering mechanisms of shallow landslides, Nat. Hazards Earth Syst. Sci., 8, 1149–1159.

Simoni, S., Zanotti, F., Bertoldi, G., and Rigon, R. (2008) Modelling the probability of occurrence of shallow landslides and channelized debris flows using GEOtop-FS, Hydrol. Process., 22, 532–545

Vanapalli, S. K., Fredlund, D. G., Pufahl, D. E., and Clifton, A. W.(1996) Model for the prediction of shear strength with respect to soil suction, Can. Geotech. J., 33, 379–392

7. Distribution of soil and mechanical parameters are assumed triangular and then distributions of FS are estimated by means Monte Carlo approach. The approach is fine but clearly it increases the computational effort. Other approaches to estimate probability of FS have been proposed in the literature. For example, the First-Order Second Moment (FOSM) (Benjamin and Cornell, 1970) is commonly used to estimate analytical approximations of the spatio-temporal FS statistics (i.e. mean and variance), that can be used to fit a theoretical probability distribution for FS and estimate the spatio-temporal dynamics of probability of failure. Moreover, mechanical parameters are normally assumed to be described by the Normal distribution (Abbaszadeh et al., 2011; you can refer to Arnone et al., 2016 and references therein). Please, briefly discuss.

Abbaszadeh M, Shahriar K, Sharifzadeh M, Heydari M. 2011. Uncertainty and reliability analysis applied to slope stability: a case study from Sungun copper mine.

Geotechnical and Geological Engineering 29: 581–596.

Arnone E, Dialynas YG, Noto LV, Bras RL (2016) Accounting for soils parameter uncertainty in a physically-based and distributed approach for rainfall-triggered landslides. Hydrol Process 30:927–944

8. Model application is a bit confusing. My impression is that it is mainly addressed to demonstrate the model capabilities instead of producing reliable landslide hazard maps for the study areas (AUC are low and FS parameters are not site-dependent). Please, state clearly the main targets of the model application.

Please, read in the following my specific comments.

1. P9L6: I suggest moving this info (soil density) in the model application section. Please, specify what type of soil density this value accounts for (total, dry, wet, bulk density...)

2. P9L20: how do you justify this low resolution of the hydrological model? Clearly, this is not able to simulate the 'local' moisture dynamics at hillslope scale . . .

3. P15L7: working resolution is 30 m. However, if I understood well some components of the system (e.g. VIC) work at coarser resolution . . . Is any interpolation method being used?

4. P16Table16: maximum values of friction angle seem to be very high . . . Do you have references?

5. P16L9: The estimation of root cohesion belong to a further 'branch' of scientific literature of this field which here seems to be significantly simplified (e.g. Pollen and Simon 2005; Preti et al., 2010; Schwarz et al., 2013 ). . .

Schwarz, M., F. Giadrossich, and D. Cohen (2013), Modeling root reinforcement using a root-failure Weibull survival function, Hydrol. Earth Syst. Sci., 17, 4367–4377.

Pollen, N., and A. Simon (2005), Estimating the mechanical effects of riparian vegetation on stream bank stability using a fiber bundle model, Water Resour. Res., 41, W07025

6. P19sec3.2.2: please give some synthesis of the characteristics of the hydro-climatology forcing for the area (e.g. some characteristic time series ... ).

7. Figure 4: how about the map of soil evolution model?

8. Figure 5: which soil properties did you use for this figure? I don't see much difference in concavity between zone (2) and zone (3). I suggest adding the degree axis in y, slope is not easily readable. Please, specify what the angle values stand for.

9. P20L7: please, specify the color of the dot lines.

10. P21L15: why is it tan(theta)$< \frac{1}{2}$ tan(phi) .... From eq. (1) it should be simply tan(theta)<tan(phi).

11. P21L18: I don't see where theta=17 degree is in the figure 5

12. P22L5: specify color of the lines?

13. Figure 6a,c: Relative frequency is in time or space?

14. Figure 6b,d: consider to cut the FS values ad significantly 'stable' values, e.g. > 10 ( no matter if Fs is 10 or 200)! Otherwise make FS in logarithm scale (interesting values are those close to 1).

15. Figure 6: Interesting questions here could be: which is the soil depth which causes a 'critical change in FS, i.e. that lead the FS going from stability to instability. And in which time window this is reached?

16. Figure 12: make figure 5 and figure 12 consistent to facilitate the comparison.

17. Please, note that the obtained values of AUC are very low ... Are you able to identify which landslides are you missing?

18. Figure 15: I suggest reporting the AUC values of the ROC curves.

---

## Author Response (AR1)

Title: A hydro-climatological approach to predicting regional landslide probability using Landlab
Author(s): Ronda Strauch et al.
MS No.: esurf-2017-39

**Response to Anonymous Referee #1**

| No. | Comment | Response |
|---|---|---|
| 1 | Main Problems:
1. Manuscript structure: the paper is long and a little bit convoluted. In my opinion the length of the paper could be reduced without any important loss of information and the structure could be improved. For example the sections related to the cyberinfra-structure Landlab (sect. 2.2 and figure 3 and the last part of the section 2.3) could be reduced or removed since it is less important for the reader of ESurf (see Aims and Scope of the ESurf Journal). Some parts are difficult to understand (see other comments) and there are some repetitions that can be removed. | We feel that the manuscript supports the aims and scope of eSurf, such as "numerical modelling of Earth surface processes." The model is also built to work with Landlab, which was described earlier this year in eSurf:
    Hobley, et al., Earth Surf. Dynam., 5(1): 21-46, 2017.
We have reduced the length of the manuscript by deleting material described elsewhere in cited papers (e.g., soil evolution component) or the software User Manual. Reduced description of hydrologic data processing in Section 2.3. *Removed repetitions* and details about Landlab in Section 2.2. However, additional material was added to address referee comments.

We choose to retain Fig. 3 because it provides visual context of the study area and mapped landslides for readers unfamiliar with the area. |
| 2 | Basic Assumptions:
2a. The authors fixed a soil density equal to 2000 kg/cubic meter constant over the entire domain; is this relative to the bulk density of the soil or to the wet soil density? Is this assumption realistic considering that you have different soil type in your domain? | Soil density represents saturated bulk density of soil (as stated in the paper), which is the same as wet bulk density. The study area has two similar soil types: sandy loam and loamy sand. A saturated bulk density of $2000 \ km/m^3$ has been used in other similar models and studies as a constant (e.g., Shalstab by Montegomery and Dietrich 1994 and SINMAP by Pack et al. 1992). However, model flexibility allows users to provide varying values of soil density and associated uncertainty as a distributed field throughout a study domain as constrained by available data.
In addition, the infinite slope stability model has been found to be insensitive to soil unit weight (density*g) (Hammond et al. (1992) and Lepore et al. 2013); *text added Page 16, lines 1-2.* |

| 3 | 2b. The authors assume the soil as incoherent (C=0) assigning all the cohesion to the root? Again is this assumption realistic? Please consider that also a loamy sand could provide a cohesion greater than that given by the root system. Please try to justify this assumption using field data relative to the soil mechanics parameters. | The referee is correct that we considered soils cohesionless. Generally soil behaves much like cohesionless soil when the clay fraction is <15% (Kulhawy et al. EPRI – Manual on Estimating Soil Properties of for Foundation Design, EPRI EL-6800, 1990). Our soils have less than 10% clay and thus low in cohesion (DOA-NRCS, 2012). The aim of our model is for risk assessment; thus, we use total cohesion given the assumption that soil cohesion is a small fraction of the total cohesion. *Added clarification on page 14, lines 7-11.* Our aim was to develop a model for regional applications that can utilize existing spatial information; thus, no field data is collected. |
|---|---|---|
| 4 | 2c. The authors assume that the recharge is given by the sum of the baseflow and the surface runoff (page 10) at each VIC grid cell. It is not clear the reason of such an assumption since usually the recharge is given only from the subsurface flow (i.e. part of the baseflow) as highlighted by the authors (page 6 – line 23-25). In similar modelistic approach (SINMAP) R is considered as a climatic factor (rainfall). Please clarify this apparent contradiction. | In our model application, we made the assumption that in steep forested mountainous landscapes runoff is generated by saturation excess mechanism due to relatively high soil infiltration capacities. A steady-state kinematic wave approach for subsurface flow is used with a depth-averaged hydraulic conductivity. This model technically requires recharge, defined as precipitation and snowmelt less of evapotranspiration and soil storage. In the VIC model this corresponds to the sum of surface and subsurface flow rates taken and averaged over a grid cell as input recharge rate. This approach is still less conservative than using rainfall directly as recharge input to the model. The use of the annual daily maximum sum of runoff and baseflow is designed to represent when the ground is likely to be the *most* saturated. *Additional clarifying text added on page 9, lines 11-16.* |
| 5 | 2d. It is not clear how the soil depth evolution model and the stability model are coupled (if they are coupled). I thought that the outcome of soil depth evolution model is provided as soil depth map (in terms of mode) but there is a sentence (page 11 – line 26-27) which is in contrast with my previous thoughts | The soil evolution model is not coupled to the probabilistic stability model and the reviewer is correct that the outcome of the soil evolution model did produce a soil depth map used as the mode in running the stability model. This map is developed as an alternative to SSURGO to better capture the spatial granularity of soil depth due to topography. The model is also used to obtain uncertainties in soil depth due to |

| | | |
|---|---|---|
| | ("Eq (a) and Eq (2) are used to calculate FS within the soil evolution model". So please try to clarify the connection between these models. I think that a figure with a flow chart describing models and connections together with the setup of the experiment could be useful to the readers. How many simulations did they run? | temporal fluctuations as a result of episodic landslides such that this uncertainty can be used to parameterize a probability density function. The soil evolution model includes a infinite slope stability model within it, to represent the long-term effects of landslides on the temporal variability of soil thickness as the model iterates over many years to produce a long-term soil depth record. We now indicated this purpose clearly.

*Sections 2.4. and 4.1.2. edited in detail* in response to referee's comments. Edits clarify the soil evolution model, how it was implemented and how the model results are used in the LandslideProbability component.

The soil evolution model was run for *10,000 years (line 3 on page 20)*. Based on the statistics obtained from the soil evolution model, the infinite slope model Monte Carlo simulation was run for *3,000 iterations (line 8 on page 23)*. |
| 6 | 3. Choice of model parameters: the choice of geotechnical and soil parameters (mode and range of variability used for MonteCarlo simulation) is, at least, not convincing.
a. The internal friction angles are fixed in Table 1 in terms of mode, min and max. I'm not convinced by these values; they seems to be very high especially for the loamy sand and sandy loam. Could you provide references or field data used to fix these values?
b. The authors use different relationship to define minimum and maximum value of trasmissivity T and friction angle. How do they define these relationships? What is the impact of these values on the final results (sensitivity analysis). | Section 3.2.1 describes the parameterization of vegetation and soil properties. The source of internal friction angle mode values shown in Table 1 are *described in line 42 on page 15,* specifically Table 5.5 within Hammond et al., 1992 and Table 5.2 within Shelby, 1993. The values are corroborated by online sources such as: http://www.geotechdata.info/parameter/angle-of-friction.html.

The minimum and maximum relationships to mode for friction angle were determined by the minimum and maximum values found in the literature and requisite right-skewed distribution. The mean, minimum, and maximum values for T shown in Table 1 are for the distributed T values over the study area and not based on the parameterized relationship for T derived as a function of mode (*see table footnote*). The parameterized relationship for T |

| | | was kept consistent with soil depth (hs) because T is partially derived from hs, along with Ksat. At the regional scale, no specific field data is collected for this study and all model parameters are obtained from the existing literature and digital maps, as the intent was to develop a regional scale model applicable with existing information. A sensitivity analysis was not included in this research as these analyses have been included elsewhere (Hammond et al., 1992; Sidle 1984); *clarified on page 6, lines 26-27.* |
|---|---|---|
| 7 | 4. Low performance of hazard maps: the authors affirm that the performance of pro- posed approach is modest. I agree with them and if the aim of this model is to create a map of landslide hazard, better results could be achieved using classical susceptibility approach based on statistical methods or data-driven methods. Moreover I think that they can remove the CD approach to test the performance of the proposed approach. The CD approach is aimed to highlight the existence of a statistically significant difference between the two P(F) cdfs (within and outside) for fixed soil depth scheme. The authors can only affirm that the two cdfs are different but this does not imply that the model performances are acceptable. I understand that this could be a first level check, but I think that can be removed without any problem for the paper. | The objective of this paper was stated in the second paragraph of Introduction, which did not include the use of a statistical model. Classical susceptibility using statistical methods may 'perform better' against observed landslides than a physical model.  However, this is because statistical approaches are based on the observations while physical models are not, which are then compared to the model.  Part of the purpose for the Landlab landslide component is to base landslide prediction on physical processes that allow for prediction (1) in areas *without* observed landslides and (2) under conditions (climate and vegetation) that may change in the future.  Statistical susceptibility models suffer from the assumption that the statistically derived relationships between historical landslides and site conditions at the time of the inventory holding true in the future. Challenges in validating physical models with observations are described in lines 28-37 on page 33.

*The CD comparison was removed* from the paper as suggested by the reviewer. |
| 8 | *Other comments:* Page 7, lines 9-10: The sentence is not clear. How does the use of maximum annual daily recharge help to define uncertainty in R? | *Clarified sentence Page 6, lines 13.*  The dataset developed from maximum annual daily recharge was used to represent the uncertainty of R over time at each grid cell. |

| 9 | Page 7, equation (3a): Please define n and n(FS<1). | Equation 3a has been *clarified (page 6)* to $$P(F) = P(FS \leq 1) = n(FS \leq 1)/N$$ where n() is the number of conditions met in bracket and N is the number of iterations |
|---|---|---|
| 10 | Page 9, line 9-21: The difference between options 2 (lognormal) and 3 (lognormal spatial) is not enough clear. Also the option 4 is not clear. Please clarify this paragraph. | The difference between the lognormal and lognormal-spatial options is the first one applies a lognormal distribution uniformly over the model domain, while the second applies a different lognormal distribution to each grid cell. *Additional clarification provided on page 8, lines 33-40.* |
| 11 | Page 9, line 30: what is core node? Is it a computational element of spatial domain? I think that these details on the computational framework are not necessary since they create a little bit confusion in the main line of the paper. | Nodes are an architectural feature of Landlab and represent the central point of a grid cell. Core nodes are the nodes that the modeler chooses to evaluate or perform the calculations on. We *removed the word 'core'* because it wasn't necessary and creates confusion. |
| 12 | Page 10, lines 16-19: please check the sentence since it is not very clear. | *Improved clarity to sentence within Sect. 2.3.* |
| 13 | Page 15, line 11: define combined curvature (reference). Is it different from the total curvature used further? | Combined and total curvature are the same, also referred to standard curvatures, which combines plan and profile curvatures. To avoid confusion, *'combined' was changed to 'total' within the manuscript*. |
| 14 | Page 16, line 22-23: sentence not clear; lines 23-25: this sentence can be removed. | *Paragraph was deleted.* |
| 15 | Page 17, lines 7-8: what is the meaning of "spatially consistent". You can use consis- tent when map is compared with the field data. Did you carry out this task? | *Sentence (page 15, lines 10-11) was revised to clarify* that the evolved soil map is more spatially heterogeneous than the SSURGO soils map with map unit polygons |
| 16 | Page 17, lines 18-19: how can the authors "confirm" the ranges of soil depth used through a long term evolution model which needs to be calibrated on soil data as well? | The soil evolution model was used to confirm the soil skewed distribution used in the triangle distribution. *Sentenced revised accordingly on page 15, lines 20-22.* |
| 17 | Page 19, lines 15-17: how the authors calculate pore-water pressure starting from the maximum daily recharge? Do they use pore- | R is an input to calculate pore-water pressure in the model, which is intertwined in the formulation and cannot be immediately shown. |

| | water pressure in their stability model? I think they use directly the recharge (see equation 2). | |
|---|---|---|
| 18 | The section 4.1.2 is not clear especially the role of regression based equation which seems to provide the soil depth as a function of slope and curvature. If you run the soil evolution model, why do you need a regression to obtain soil depth?

In the same section it is not clear the difference between M-SD and M-SD LT. I think that there is a lot of information but this is not well-organized. | *We improved the description and application* of the soil evolution model and how we used it. Section 2.4. describes broadly why we needed a soil evolution model, what we obtained from it, and how we used the modeled soil depth. It also provides a narrative of the soil production processes modeled.

*Section 4.1.2. describes in more detail* how the model is implemented at select locations that represent the topography and vegetation of the domain and how this limited model information is used to make a map of mode of soil depth and set minimum and maximum parameters of the triangular distributions used in the Landlab Landslide Probability component. This section clarifies the differences between M-SD and M-SD LT. |
| 19 | Figures 10 and 11 highlight the same information (Probability or return period). Please consider removing one of the two figures. | We would like to *retain both these figures* because, while they highlight similar albeit transformed data, they communicate different information about landslide hazard. This may be meaningful for different readers, especially resource managers who may prefer return period information. |
| 20 | Page 32, line 13: The authors are not using "observations" in figure 13 but model results. Please change the sentence. | Correct, *changed* "observations" to 'model results' on *page 31, line 5*. |
| 21 | Figure 13c: in the legend line relative to M-SD LT is missing. | The mean soil depth line for M-SD LT was *purposely left out* because it is relatively similar to M-SD and we desired minimizing the lines shown in the figure. |
| 22 | Page 34, line 16: since the authors use 10% of highest elevation cell, I suggest to remove 20% and 30%. | Suggestion accepted and 20% and 30% *reference removed*. |
| 23 | Page 34, lines 24-26: I suggest specifying the number of DA source cells and the number of DA outside source. | We specified the number of DA outside source cells as 50,000 sample. The number of source cells was *added at 4318 grid cells on page 31, line 31*. |

| 24 | Figure 14a: I think there is an error in the plot. The sum of all the bars must be equal to 1. If this is true for Outside DA, it cannot be true for Source DA since for each bin the relative frequency is lower. | This *figure has been removed*. |

**Response to Anonymous Referee #2**

| | | Comment | Response |
|---|---|---|---|
| 25 | The work is of high scientific value and the applied methodologies are scientifically robust. However, to my opinion, the paper ended up being excessively long, sometimes not immediately clear and the overall application not well focused on clear and simple targets. Moreover, given the numerous details of the developed numerical model system, I wonder if a journal which addresses to models development or environmental software would be more appropriate. Anyways, to my opinion, it needs major revision for it to be published in order to make it clearer, more fluent, to better define the aims of the application and to improve the literature review which lacks of some important contributions. | Please refer to response to referee #1 comment #1. Additional literature review with citation has been added, particularly based on the additional citations supplied by the referees.

We also made the effort to improve clarity of the objectives of the paper, the purpose of different simulations, findings with respect to model results, and related observational inferences. |
| 26 | 1. The manuscript is very long and sometimes repetitive, with English style a bit verbose. I have to read it a couple of times to get to the point. Some parts can be synthesized and stated more directly. | We agree that the article is long, but feel it reflex the necessary information to: (1) orient the reader, (2) describe the model framework and cyberinfrastructure designed for reproducibility, and (3) demonstrate a real-world application. Nevertheless, we reduced the length of the paper and made it more direct to point. Please refer to response to referee #1 comment #1. |
| 27 | 2. Literature review is well done and comprehensive of various aspects involved in this work. However, it | Additional literature has been cited throughout the manuscript, in particular some of the literatures noted by referees. *Thirteen* |

| | | |
|---|---|---|
| | also lacks of some important contributions in the specific field of physically based modeling for rainfall-triggered landslides, also with regard with the parameters uncertainty. | *additional citations* have been added to the references. |
| 28 | 3. My main concern is whether simpler and more computationally efficient statistical approaches for susceptibility evaluation could be more appropriate for such regional and long term analysis.

The methodology is based on the use of various simplified models which make it complex. The ultimate model performances are not very satisfactory in terms of ROC and AUC. The approach is classified as dynamic and processed- oriented; however it is not able to reproduce and simulate specific events due to the simplifications and the large temporal scale and can be used only for long term analysis, thus becoming a kind of 'static' approach. Statistical models are very robust and able to guarantee very satisfactory results (e.g. Lepore et al., 2012; Lee and Pradhan, 2007).

Lepore C, SA Kamal, P Shanahan, RL Bras (2012). Rainfall-induced landslide susceptibility zonation of Puerto Rico. Environmental Earth Sciences 66 (6), 1667-1681

Lee, S., Pradhan, B. (2007). Landslide hazard mapping at Selangor, Malaysia using frequency ratio and logistic regression models. Landslides 4, 33-41 | We are familiar with the literature provided and agree that statistical analysis is a powerful modeling technique for landslide hazard assessments for existing conditions. However, the aim of our research is different, as clearly indicated in the Introduction section. Statistical models are particularly useful for evaluating conditions conducive to different types of landslides.  Indeed, some of the authors are completing a manuscript that compares a statistical approach to the physical model used in this paper, which will be submitted for peer review in the coming months. See also response to referee #1, comment #7.

Our model can be used at event scales in addition to long term. When using for a particular event, the user would need to quantify the uncertainty for recharge and other soil parameters to reflect the conditions at the time of the event. The Landlab LandslideProbability component is based on a simple physical model and its lack of complexity facilitates its use in a wide variety of settings and situations.

We agree that the performance against mapped debris avalanches with the physical model is modest based on ROC and AUC metrics, as stated on line 19 on page 32.  Challenges in validating physical models with observations are described in lines 28-37, page 33.  We believe the modest performance based on observed landslides does not discount the value of a predictive physical model, particularly where data are limited. |
| 29 | 4. In the model system there is a mix of temporal and spatial resolutions | Yes, there is a mix of temporal and spatial resolutions used in the demonstration of the |

| | | |
|---|---|---|
| | (soil depth evolution at yearly scale, hydrological model at daily and 6 kmq square, geomorphological model at 30 m). If I understood well the finest temporal resolution is the daily scale of the annual maximum recharge. However, the daily resolution misses the most intense events and moreover, the daily annual maximum recharge does not guarantee the worst 'hydrological conditions' since the antecedent soil moisture conditions are also influent. This why I am skeptical on the advantage of this approach instead of others (comment 3). | model. The hydrological model is a daily time-step, but only the annual maximum is used in the application. We agree that the daily hydrology time step misses the sub-daily intense storms, which may lead to some underestimate of instability. However, landslides driven by pore-water pressure as subsurface flow develops requires longer durations than short storm outbursts. The subsurface flow model is a steady-state model so no antecedent moisture conditions is required in the model. It assumes that the subsurface flow attains steady-state given the annual maximum recharge value.

The spatial scale variety used in the model application is a result of the native resolution of the various data sources as well as the resolution chosen for the application. However, the model is flexible for use at other spatial and temporal resolutions. |
| 30 | 5. Soil depth evolution: it is not totally clear to me whether the soil depth evolution model is run in conjunction with the stability module or it is run 'off line' and the final map is then fed to the slope stability module.

Also, how the soil depth evolution influences the hydrological module? Theoretically, the soil evolution model itself should take into account the change in elevation due to the landslide. Is this done? Please make it clear. | We improved the description and application of the soil evolution model and how we used it. Section 2.4. describes broadly why we needed a soil evolution model, what we obtained from it, and how we used the modeled soil depth. It also provides a narrative of the soil production processes modeled.

*Section 4.1.2. describes in more detail* how the model is implemented at select locations that represent the topography and vegetation of the domain and how this limited model information is used to make a map of mode of soil depth and set minimum and maximum parameters of the triangular distributions used in the Landlab LandslideProbability component. This section also clarifies the differences between M-SD and M-SD LT.

The soil evolution model does not influence the hydrology model and landscape elevations are not changed with soil development and landslides. |

| 31 | 6. Authors do not explicitly discuss the importance of the effect of matric suction and the 'apparent' cohesion which arises under unsaturated soil moisture conditions (e.g., Simoni et al., 2008; Baum et al., 2002) and which can be much higher than soil and also root cohesion (e.g., Arnone et al., 2016). They discuss clearly hypothesis of steady state conditions, but this does not justify the neglecting of the matric suction. Moreover, several procedure have been also proposed to predict shear strength under unsaturated soil (based on modified Mohr-Coulomb failure criterion (Vanapalli et al., 1996; Fredlund et al., 1996)), which have been used in various works (Montrasio and Valentino, 2008; Lepore et al., 2013). I suggest referring to Lepore et al., (2013) for a discussion on this point.

Arnone E, Caracciolo D, Noto LV, Preti F, Bras RL (2016) Modeling the hydrological and mechanical effect of roots on shallow landslides. Water Resour Res 52(11):8590–8612

Baum, R. L., Savage, W. Z., and Godt, J. W. (2008) TRIGR-a Fortran program for transient rainfall infiltration and grid-based regional slope-stability analysis, US Geological Survey Open File Report 2008-1159, 75 pp.

Fredlund, D. G., Xing, A., and Barbour, M. D.(1996). The relationship of the unsaturated soil shear strength to the soil water | We recognize and explicitly state that we neglect apparent cohesion on line 41 on page 33. In steep mountain terrain of the Pacific Northwest soils, are loosely developed and have large particles in hillslope soil mixtures. Earlier applications of similar model typically use cohesionless soils. However, we recognize the discoveries of the importance of matric suction, perhaps more important in well-developed soils, in stability analysis (e.g., citations referee listed) and highlight this opportunity for future advancements in the Landlab LandslideProbability model on page 34, line 5-6. We believe that comparison of our model with the tRIBS-VEGGIE model would be an interesting future study, but is beyond the scope of the current model version.

We are familiar with many of the citations provide and appreciate exposure to the others. Some of these citations are already included in the manuscript, but the *literature review has been enhanced (13 more citations)* using these references as well as others. *Additional or modification of text from these references include: insensitivity to soil unit weight (Page 16, line 12), assumed negligible correlation between C and friction angle (Page 8, line 14), and hydrologic effect of roots drying out soils (Page 31, lines 15-17).* |

| | | |
|---|---|---|
| | characteristic curve, Can. Geotech. J., 32, 440–448 | |
| | Lepore C, Arnone E, Noto LV, Sivandran G, Bras RL. (2013). Physically based modeling of rainfall-triggered landslides: a case study in the Luquillo forest, Puerto Rico. Hydrology and Earth System Sciences 17: 3371–3387. DOI: 10.5194/hess-17-3371- 2013. | |
| | Montrasio, L. and Valentino, R. (2008) A model for triggering mechanisms of shallow landslides, Nat. Hazards Earth Syst. Sci., 8, 1149–1159. | |
| | Simoni, S., Zanotti, F., Bertoldi, G., and Rigon, R. (2008) Modelling the probability of occurrence of shallow landslides and channelized debris flows using GEOtop-FS, Hydrol. Process., 22, 532–545 | |
| | Vanapalli, S. K., Fredlund, D. G., Pufahl, D. E., and Clifton, A. W.(1996) Model for the prediction of shear strength with respect to soil suction, Can. Geotech. J., 33, 379–392 | |
| 32 | 7. Distribution of soil and mechanical parameters are assumed triangular and then distributions of FS are estimated by means Monte Carlo approach. The approach is fine but clearly it increases the computational effort. Other approaches to estimate probability of FS have been proposed in the literature. For example, the First-Order Second Moment (FOSM) (Benjamin and Cornell, 1970) is commonly used to estimate analytical approximations of the | FOSM approach provides the moments of a random function.  Given these moments, then a theoretical distribution of FS would need to be assumed at each grid cell.  Monte Carlo simulation does not assume any distribution, but estimates probability of failure based on the calculated FS values <=1 from the distribution of FS given the distributions of input variables. Malkawi et al. (2000) compared FOSM and Monte Carlo simulation of slope stability and found good agreement between the two approaches using 2 to 3 slope stability methods; slight differences between the two approaches for the Spencer method was due to the need to |

| | | |
|---|---|---|
| | spatio-temporal FS statistics (i.e. mean and variance), that can be used to fit a theoretical probability distribution for FS and estimate the spatio-temporal dynamics of probability of failure.

Moreover, mechanical parameters are normally assumed to be described by the Normal distribution (Abbaszadeh et al., 2011; you can refer to Arnone et al., 2016 and references therein). Please, briefly discuss.

Abbaszadeh M, Shahriar K, Sharifzadeh M, Heydari M. 2011. Uncertainty and re- liability analysis applied to slope stability: a case study from Sungun copper mine. Geotechnical and Geological Engineering 29: 581–596.

Arnone E, Dialynas YG, Noto LV, Bras RL (2016) Accounting for soils parameter un- certainty in a physically-based and distributed approach for rainfall-triggered landslides. Hydrol Process 30:927–944 | perform a numerical approximation of the first derivative of factor-of-safety required by FOSM, which isn't required using Monte Carlo simulation.  They endorse the use of the Monte Carlo simulation approach given the capabilities of computers to handle data and computation. We did not find computational requirements limiting in our use of Monte Carlo simulation.  In fact, once the model input is prepared, the model's Monte Carlo simulation with n=3,000 for an area of over 2,700 km2 runs in minutes.

Malkawi, Abdallah I. Husein, Waleed F. Hassan, and Fayez A. Abdulla. "Uncertainty and reliability analysis applied to slope stability." *Structural safety* 22.2 (2000): 161-187.

Regarding distributions of parameters, both normal and uniform distributions are often used in slope stability analysis, but we preferred to use triangle for *reasons given on page 8, lines 3-5.*  This distribution has been used by others in slope stability modeling using Monte Carlo simulations (Cho, 2007; Dou et al., 2014; Hammond et al., 1992; El-Ramly et al., 2002; Strenk, 2010).  Additionally, it gives the most weight to data and/or knowledge from the modeler in the form of mode, facilitates skewed distributions, and avoids extrapolations to extreme high or low (and negative) values. Observed/measured values of soil depth, cohesion, and friction angle are typically skewed and not normally distributed (Hammond et al. 1992). |
| 33 | 8. Model application is a bit confusing. My impression is that it is mainly addressed to demonstrate the model capabilities instead of producing reliable landslide hazard maps for the study areas (AUC are low and FS parameters are not site-dependent). Please, state clearly the main targets of the model application. | The model was implemented at a national park to provide demonstration of the model and provide a stability analysis for the park.  It was not to substantiate the current landslide inventory. Explicit targets of the application are provided with a *new sentence at lines 36-38, page 10.* |

| 34 | 1. P9L6: I suggest moving this info (soil density) in the model application section. Please, specify what type of soil density this value accounts for (total, dry, wet, bulk density...) | Soil density is wet bulk density (see response to comment #2 of Referee #1. This *sentence was removed* as the value is listed in line 43 of page 15. |
|---|---|---|
| 35 | 2. P9L20: how do you justify this low resolution of the hydrological model? Clearly, this is not able to simulate the 'local' moisture dynamics at hillslope scale . . . | The VIC hydrological model was used to obtain annual maximum daily recharge averaged over the upslope contributing area of each grid cell of the probabilistic landslide initiation model. Local relative wetness used in the infinite slope stability model was calculated from a steady-state subsurface flow model, which is a function of local slope and transmissivity at the resolution of the landslide model. The steady-state subsurface model does not consider the local soil moisture dynamics. Additional justification for using VIC in our application is *provided in Sect. 3.2.2 on page 17*. |
| 36 | 3. P15L7: working resolution is 30 m. However, if I understood well some components of the system (e.g. VIC) work at coarser resolution ... Is any interpolation method being used? | Recharge at the 30-m grid resolution is provided by routing the upstream fractional area of the coarse 1/16° resolution VIC grid cells to calculate the upstream proportionally-averaged maximum recharge for each year. These time series are used in an interpolation step within the component to generate a cumulative distribution of recharge equal to the length of the number of Monte Carlo iterations used in the simulation. This is *clarified with revisions to page 8, lines 37-40.* More detail on this is also provided in the User Manual as mentioned on line 23, page 9. |
| 37 | 4. P16Table1: maximum values of friction angle seem to be very high . . . Do you have references? | Source for friction angle is provided in line 42, page 15; highest values are based on the ranges in the literature. See also *response to comment # 6* of Referee #1. |
| 38 | 5. P16L9: The estimation of root cohesion belong to a further 'branch' of scientific literature of this field which here seems to be significantly simplified (e.g. Pollen and Simon 2005; Preti et al., 2010; Schwarz et al., 2013 ). . . | We agree that estimating root cohesion is a 'branch' of research by itself and represents one of the more challenging variables to represent in a landslide model. Our landslide research aims for regional application and was not focused on this branch of study. We believe basing root cohesion estimates on vegetation and used in a Monte Carlo simulation is a |

| | | |
|---|---|---|
| | Schwarz, M., F. Giadrossich, and D. Cohen (2013), Modeling root reinforcement using a root-failure Weibull survival function, Hydrol. Earth Syst. Sci., 17, 4367–4377.

Pollen, N., and A. Simon (2005), Estimating the mechanical effects of riparian vegetation on stream bank stability using a fiber bundle model, Water Resour. Res., 41, W07025 | straightforward approach that can be easily implemented in other locations; however, the model is capable of operating with detailed estimates of root cohesion provided by the user.

Citations provided offer approaches for estimating root cohesion based on quantification or parameterization of detailed root strength that is beyond the regional approach we take in our model. However, recognition of these techniques was *added to the paper for exposure to interested readers on page 14, line 11*. |
| 39 | 6. P19sec3.2.2: please give some synthesis of the characteristics of the hydro-climatology forcing for the area (e.g. some characteristic time series ... ). | The seasonality and range of precipitation is discussed on page 11, lines 6-13 and mean annual precipitation is spatially depicted in Fig. 2c, which was *moved up a page*. *Additional characterization of recharge was provided in a sentence on page 17, lines 14-16.* |
| 40 | 7. Figure 4: how about the map of soil evolution model? | A map of a portion (close up) of the evolved soil depth product is in Fig. 7; however, the *Fig. 7 was revised to match the color palette of Fig. 4*'s SSURGO soil depth map. A relative histogram of a spatial soil depth product is also provided in same figure. We believe this information is sufficient to provide characterization for the comparison with the SSURGO soil depth product. Given the sample topography, details of the influence of converging and diverging morphologies on soil depth can be more clearly seen in a close-up figure than in the map of the entire domain. |
| 41 | 8. Figure 5: which soil properties did you use for this figure? I don't see much difference in concavity between zone (2) and zone (3). I suggest adding the degree axis in y, slope is not easily readable. Please, specify what the angle values stand for. | Soil properties are *listed in figure caption*, but include friction angle of 34 degrees and dimensionless cohesion values listed in legend. The break in concavity between zone 2 and 3 is subtle, but was guided by the source cells (triangles) and the saturation line. We prefer to retain slope as m/m rather than degrees to facilitate comparison with other slope-area geomorphic analyses such as Montgomery and Dietrich (1994) and Pack et al. (1998). However, |

| | | the labeled angles in degrees on the plot should help orient the reader. Assuming the angle values referee is referring to are 17, 35, and 50 degrees. |
|---|---|---|
| 42 | 9. P20L7: please, specify the color of the dot lines. | We believe this refers to Fig. 5. *Caption amended to include color of vertical lines.* |
| 43 | 10. P21L15: why is it tan(theta)< ½ tan(phi) ....From eq. (1) it should be simply tan(theta)<tan(phi). | Solving the FS equation (Eq. 1a) for $\tan(\theta)$, given FS=1, relative wetness=1, cohesionless soil, and a water to soil density ratio= ½, yields $\tan(\theta) \leq \frac{1}{2} \tan(\phi)$ for unconditionally stable conditions. |
| 44 | 11. P21L18: I don't see where theta=17 degree is in the figure 5 | $\theta=17°$ in Fig. 5 is noted next to the cyan line. Updated figure caption *based on comment #41* eases the visibility of this threshold. |
| 45 | 12. P22L5: specify color of the lines? | *Colors added.* |
| 46 | 13. Figure 6a,c: Relative frequency is in time or space? | This is in time. *Clarified in Fig. 6 caption.* |
| 47 | 14. Figure 6b,d: consider to cut the FS values ad significantly 'stable' values, e.g. > 10 ( no matter if Fs is 10 or 200)! Otherwise make FS in logarithm scale (interesting values are those close to 1). | Incorporated referee's comment and *converted FS ($2^{nd}$ Y-axis) to logarithm scale* to emphasize the values closer to 1. |
| 48 | 15. Figure 6: Interesting questions here could be: which is the soil depth which causes a 'critical change in FS, i.e. that lead the FS going from stability to instability. And in which time window this is reached? | The referee presents interesting questions that could be addressed in the soil evolution model. However, the soil evolution model is not the focus of our research, but provides a mechanism to estimate soil depth to compare with soil surveys or to use in areas lacking soil depth estimates. |
| 49 | 16. Figure 12: make figure 5 and figure 12 consistent to facilitate the comparison. | *Partially adjusted Fig. 5* to reflect the same maximum Y-axis limit as Fig. 12 *at $10^1$ and improved legend.* Otherwise retained the figure limits to maximize the display of data within the axes limits. |
| 50 | 17. Please, note that the obtained values of AUC are very low ... Are you able to identify which landslides are you missing? | We recognize and report that the AUC values indicate modest model performance with observed debris avalanches on page 32, line 19. We have examined the landslides that the model did not identified as high probability as well as areas not mapped as landslides, but with high probabilities. We have not identified a consistent pattern in apparent "mis-matches". |

| | | Additional evaluation of these areas is noted as future work on page 33, lines 24-26.
For more discussion about performance, *see comment #7.* |
|----|----|----|
| 51 | 18. Figure 15: I suggest reporting the AUC values of the ROC curves. | We report the AUC values in the text on page 32, line 20; however, we *repeated the range in the Fig. 14 caption.* |

[revised manuscript text omitted]

**2 Methodology**

2.1 Probabilistic approach to landslide initiation

The infinite slope stability equation, derived from the Mohr-Coulomb failure law,  predicts the factor-of-safety (FS)  of an infinite plane from the ratio of stabilizing forces of  cohesion and friction, reduced by pore-water pressure , to destabilizing forces of gravity (Hammond et al., 1992; Wu and Sidle, 1995). The model as given by Pack et al. (1998) is:

$$= \frac{(\quad + \quad)/h}{\sin} + \frac{\cos \ \tan \ (1- \quad / \quad)}{\sin} \qquad (1a)$$

$$* = (\quad + \quad)/h \qquad (1b)$$

C* is a dimensionless cohesion (Eq. 1b) embodying the relative contribution of cohesive forces to slope stability. When C*>1, cohesion is sufficient to hold the soil slab vertically (Pack et al., 1998). $Cr$ and $Cs$ are root and soil cohesion respectively [Pa], $h_s$ is the soil depth perpendicular to slope [m], $\rho_s$ and $\rho_w$ are saturated soil bulk density and water density [kg/m$^3$], respectively, $g$ is acceleration due to gravity [m/s$^2$], $\theta$ is slope angle of the ground, and $\emptyset$ is soil internal friction angle [°]. Relative wetness, $R_w$, is defined as the ratio of subsurface flow depth, $h_w$, flowing parallel to the soil surface, to $h_s$. Deterministically, a hillslope element is unstable if $FS < 1$ and stable if $FS > 1$ (Sidle and Ochiai, 2006; Shelby, 1993). When FS = 1, the slope is "just-stable" or in a state of "limited equilibrium" (Lu and Godt, 2013).

Relative wetness is arguably the most dynamic factor at short time scales, relating to water table depth and to recharge rate. Considering that hillslope hydrology is more likely to attain equilibrium conditions during prolonged wet conditions (e.g., Barling et al., 1994; Borga et al., 2002), a steady-state representation of subsurface flow is used. It is derived from local subsurface lateral flow, $q_s$ [m$^2$ d$^{-1}$], represented by a 1-D (i.e., flow parallel to bedrock) form of

the kinematic wave approximated by Darcy's law using topographic gradient of hillslope, $q_s=K_sh_w\sin\theta$ (Wu and Sidle, 1995). Under a steady-state assumption, lateral flow is in balance with the rate of water input, $q_r$ [m$^2$ d$^{-1}$], through a uniform rate of recharge, R [m d$^{-1}$], defined across the upslope specific contributing area, $a$ [m], $q_r=Ra$. This assumption gives: $Ra=K_sh_w\sin\theta$,

5 where $K_s$ is saturated hydraulic conductivity [m d$^{-1}$]. Solving this equation for $h_w$ and dividing both sides by $h_s$ gives $R_w$ (Montgomery and Dietrich, 1994; Pack et al., 1998):

$$= \frac{h}{h} = \min\left(\frac{}{\sin},1\right) \qquad (2)$$

Here $T$ is local soil transmissivity [m$^2$ d$^{-1}$], which is depth-integrated saturated hydraulic conductivity, $K_s$. For uniform $K_s$ within the soil profile overlying  impermeable bedrock $T=K_sh_s$.

10 Ground saturates when $R_w = 1$,  the maximum value for $R_w$.

15  (Montgomery and Dietrich, 1994). These assumptions are appropriate for  steep topography  to efficiently characterize wetness over large areas (Tarolli and Tarboton, 2006; van Westen et. al., 2006).

A Monte Carlo simulation is used with equation (1a) by assuming $R$, $T$, $C$ ($C=C_r+C_s$), $h_s$ and $\phi$ as

20 random variables represented by probability distributions (Tobutt, 1982; Hammond et al., 1992).  The uncertainty in R is  represented by  using a dataset of the maximum daily

25 recharge in each year (e.g., Benda and Dunne, 1997a; Borga et al., 2002; Istanbulluoglu et al., 2004). The model includes both spatially uniform and spatially distributed options for sampling recharge (described further in Sect. 2.3). Using sampled random variables in Eq. (1a), FS is calculated in each model iteration, i, during the simulation. Annual probability of failure P(F) and landslide return period (RP) at each grid cell are defined as (Hammond et al., 1992; Cullen

30 and Frey, 1999):

$$P(F) = P(FS \leq 1) = n(FS \leq 1)/N \qquad (3a)$$
$$RP = P(F)^{-1} \qquad (3b)$$

*

35

Comment [RS2]: We don't multiply n times FS<1. We sum up the FS<1 (count)

where n() is the number of conditions met in brackets and N is the number of iterations. Our model does not predict the size of a probable landslide at the initiation point, which can be smaller or larger than the size of a DEM grid. P(F) gives a relative propensity that a landslide could initiate within the grid cell.  if some random samples lead to a low deterministic FS, they contribute to an increase of the P(F) within that cell.  sensitivity analysis of the infinite slope stability model was shon in the literature  (see: Sidle 1984; Hammond et al., 1992).

**2.2 Model Development in Landlab**

 Landlab is a python-based earth surface modeling toolkit (landlab.github.io). It provides a grid architecture, a suite of pre-built components for modeling surface or near-surface processes, and utilities that handle data creation, management, and interoperability among process components (Tucker et al., 2016; Hobley et al., 2017; Adams et al., 2017). ~~The Landlab design allows for a "plug-and-play" style of model development, where process "components" can be coupled together in a user-customized "model driver". Each component is a set of code functions that represent an individual process; the model driver has code used to import or generate required data, execute the component or set of components used in the model, and to visualize results.  For example, once a DEM is imported as a Landlab grid instance, any Landlab component can be used with interoperable methods to attach data and perform operations. Landlabsupporting(see Sect. 6)~~.

The LandslideProbability component is written in python and implemented with a model "driver" (written as a Jupyter Notebook) using the workflow shown in Fig. 1 of the component's User Manual (See Sect. 6). The driver imports Landlab and necessary Python libraries, loads and processes data, and executes the LandslideProbability component on RasterModelGrid (RMG), which is a Landlab class for creating raster grid objects. A structured grid is generated by the RMG class that covers the model domain. The spatial model parameters and model forcing data are completed in preprocessing steps outside of Landlab. variables They  are loaded and stored on grid nodes (the central point of grid cells) of the RMG as Landlab data fields, composed of NumPy arrays

).

The LandslideProbability component is instantiated by passing four arguments: the grid, number of iterations, recharge distribution, and recharge parameters. Once the component has been instantiated, the component's method *calculate_landslide_probability*() is executed in a for loop that performs the calculations at each node. The number of iterations in the range of 700 (Malkawi et al., 2000) to >1,200 (Abbaszadeh et al., 2011) were found sufficient in the literature. We used 3,000 in this study. At each node the method generates unique model parameters, and calculates the relative wetness (Eq. 2) and FS (Eq. 1a) for each iteration. At the end of the iterations, probability of saturation and probability of failure are calculated at each node.

[Figure]

**Figure 1.** Workflow for landslide modeling using the Landlab LandslideProbability component. The user creates input parameter fields (purple box). The model driver (gray) imports Landlab, Python libraries, and model parameter fields instantiates (e.g., create an instance) the RasterModelGrid and the component; and runs utilities and the Landlab component (blue inside dashed box).

Slope angle and specific contributing area are static parameters derived from a DEM in pre-processing steps. Total cohesion, $C$ (i.e., $C_r + C_s$), $\phi$, $h_s$, and $T$ are treated as random variables following a triangular distribution specified with three parameters (minimum, mode, and

maximum)  (Cho, 2007; Dou et al., 2014). Options for user-provided $T$ or $K_s$ are accepted by the component; although comparison of resulting landslide probabilities were found to be similar given that the value of $T$ was derived from $h_s$. Triangular distributions give weight to the most likely value (i.e., mode) and have been proposed in other Monte Carlo simulation studies for slope stability (Hammond et al., 1992; El-Ramly et al., 2002; Strenk, 2010).

Mode parameters of the triangular distribution used for all soil and vegetation parameters are developed as raster grids as part of preprocessing steps, loaded to Landlab, and assigned to nodes of the RMG (Fig. 1). For root cohesion we used  vegetation types from the  National Land Cover Data (NLCD) (Jin, 2013; USGS, 2014b) , with a lookup table for cohesion obtained from the literature (Table 1). Only for cohesion, minimum and maximum parameters are also provided as raster grids to represent distributed variation with vegetation. Gridded Soil Survey Geographic Database (SSURGO) (*DOA-NRCS* 2016) is used to assign $\phi$, $h_s$, and $T$ (see 3.2.1 for details). The current model design assumes negligible correlation between C and $\phi$ as assumed in other studies (e.g., Abbaszadeh et al., 2011; Arnone et al., 2016a). Other spatial soil and vegetation datasets can be used in the preprocessing of the model. Exposed bedrock and glaciated surfaces can be excluded from the model domain by user.

In each Monte Carlo iteration, we use annual maximum daily recharge, R, which represents a steady-state uniform recharge rate defined for the upslope contributing area of each RMG node . Local recharge (i.e., flux of water entering saturated zone) within the upslope contributing area of RMG nodes can be incorporated from a variety of grid resolutions from hydrologic models, referred to as a Hydrologic Source Domain (HSD). A "Source Tracking Algorithm" (STA) is developed that uses spatially variable recharge data from a HSD, re-sampled to the grid resolution of slope stability calculations, and routes local recharge in the downstream direction following the steepest descend until a target cell is reached. Then it calculates the spatially-averaged upslope recharge for each node of the RMG, used as R in the model. STA is described in more detail in the component's User Manual (See Supplement).

Four options for sampling  R are provided for Monte Carlo simulation at each node,  identified in the model driver by selecting a probability distribution: *uniform, lognormal, lognormal_spatial,* and *data_driven_spatial*. The first two options assign spatially uniform random variables of R across the whole model domain with respective parameters of minimum and maximum, and mean and standard deviation. The latter two "spatial" options are designed to represent spatial variability in R, constructed based on the statistics of annual maximum R obtained from a HSD using the STA utility  The *lognormal_spatial* option assigns mean and standard deviation of R at each node derived from the modeled R data, while the

*data_driven_spatial* option uses a non-parametric sampling approach to sample from the cumulative distribution of R data produced for each node of the RMG. In this regional application of the landslide component, the VIC macroscale (1/16° or 5x6 km grid cell) hydrology model is used as HSD. annual maximum recharge and uses a lognormal distribution of recharge for simulation. The *data_driven_spatial* option calculates uses a non-parametric Monte Carlo sampling approach to sample directly from historical a recharge data distribution of. Uupslope-averaged recharge for each grid node is calculated with the Landlab Source Tracking Algorithm (STA) utility using recharge from a HSD, which in this study issuch as the VIC macroscale (1/16° or 5x6 km grid cell) hydrology model, and by interpolating a cumulative distribution of recharge equal to the length of the number of Monte Carlo iterations used in the simulation .

Within the model driver, the user also sets any boundary conditions, such as areas to exclude (i.e., bedrock outcrops, glaciers) and assigning the number of Monte Carlo iterations (n>>1,000, Hammond et al., 1992). The seed random number generator does not appear to affect Monte Carlo simulation results and n>700 (Malkawi et al., 2000) or n>1,200 (Abbaszadeh et al., 2011) is sufficient to converge to the same probability of failure. The LandslideProbability component is instantiated by passing four arguments: the grid, number of iterations, recharge distribution, and recharge parameters. Multiple instances of the LandslideProbability class can be established in one driver to compare the results from different recharge specifications. Once the component has been instantiated, the component's method *calculate_landslide_probability()* is run. For each iteration, this method loops through each core node, generates unique model parametervariables, and calculates the relative wetness (Eq. 2) and deterministic FS index (Eq. 1a) at each iteration. At the end of the iterations, the P(F) at the node is calculated as the number of iterations in which FS≤1 divided by the number of iterations (n). Variables output by the component at each core node include calculated probability of saturation and P(F), which can be queried at each node or visualized across the entire grid within the driver or using a command line terminal to execute commands.

**2.3 Hydrologic Data Processing**

A key aspect of the regional landslide modeling approach is the linking of landslide hazard to hydro-climatological forcing at regional scales. The Landlab LandslideProbabilty component is written with the capability to accept input from hydrologic model outputs, such as. We used the VIC macroscale hydrologic model (Liang et al., 1994) we to demonstrate in this paper. this capability because it VIC is semi-distributed, predominantly physics-based macro-scale hydrology model that characterizes elevation-dependent differences in regional precipitation and temperature forcings and their influence on recharge through regulating rain-on-snow, snow accumulation and melt, rain-on-snow, evapotranspiration, and soil moisture. VIC is semi-distributed, predominantly physics-based macro-scale hydrology model, which is advantageous for representing distributed parameters of hydro-climatology that are not stationary in time over large regional areas (Hamlet et al., 2013)..

The steady-state subsurface model coupled with the infinite slope stability equation in our model requires a steady-state recharge rate as input. Recharge refers to the input of water to subsurface flow from precipitation and snowmelt less  evapotranspiration and soil storage. In a VIC model simulation, this condition can be obtained by adding baseflow and surface runoff. Observations and model experiments suggest that widespread landslides  are usually associated with the largest rainfall events (e.g., Page et al., 1994; Gorsevski et al., 2006). To characterize  when the ground is likely to be the most saturated each year, daily baseflow and surface runoff are summed at each VIC grid cell to represent daily recharge [mm $d^{-1}$] and the annual maximum daily value is selected for each  year of the dataset, similar to others (e.g., Benda and Dunne, 1997a; Borga et al., 2002; Istanbulluoglu et al., 2004).   To obtain a steady-state average recharge rate in the upslope contributing area of each RMG,  the  Landlab STA utility is used  (see section. 2.2., and Fig 1. of User Manual link provide in Sect. 6).

2.4 Soil Evolution Model

Soil depth controls the temporal and spatial patterns of landsliding over geomorphic time scales and is considered one of the most significant variables controlling the FS stability index, especially at depths less than 1.5 m (Benda and Dunne, 1997a; Istanbulluoglu et al., 2004; Catani et al., 2010; Sidle and Ochiai, 2006). Soil depth can vary in space and time as a function of weathering and sediment transport in relation to climate, lithology, topographic position, and vegetation cover (Dietrich et al., 1995). Despite its fine grid resolution, the SSURGO database (*DOA-NRCS* 2016) only broadly captures topographic controls on soil depth and reflect existing conditions in the field based on soil surveys. In an attempt to

improve the representation of spatial granularity and local uncertainties of soil depth,  a soil evolution model is used  (Dietrich et al., 1995; Simoni et al., 2008; Pelletier and Rasmussen, 2009; Tesfa et al., 2009; Bellugi et al., 2015). The model is run to develop time series of soil depth from which triangular distribution parameters for soil depth (mode, minimum and maximum) can be obtained and used in Landlab LandslideProbability component.

In the soil evolution model, change in soil depth is modeled as the annual sum of local soil production, divergence of sediment flux due to soil creep, and soil removal by landslides (e.g., Tucker and Slingerland, 1997; Heimsath et al., 1997; Braun et al., 2001; Istanbulluoglu et al., 2004; Nicótina et al., 2011). The rate of soil production is related exponentially to local soil depth (Heimsath et al., 1997). Soil creep is linearly related to local elevation gradient (e.g., McKean et al., 1993). Soil removal by landslide initiation is modeled with the infinite slope stability equation, implemented with representative parameters (Table 2). When FS$\leq$ 1,

$$\frac{\partial z_b}{\partial t} = -P_0 e^{-\alpha h_z} \quad (5)$$

$$\nabla q_s = -K_d \nabla^2 z_F \quad (6)$$

$$\frac{\partial h_s}{\partial t} = \frac{\rho_r}{\rho_s} P_0 e^{-\alpha h_z} + K_d \nabla^2 z \quad (7)$$ ~~Variable curvature profiles, steep and planar hillslopes, and abrupt knife edge drainage divides indicate nonlinear transport processes such as mass wasting (Roering et al., 2004, 1999). These landscape characteristics are common in the steep terrain; therefore, in every iteration of the model, Eq. (1a) and Eq. (2) are used to calculate FS within the soil evolution model. This deterministic FS is independent of the FS values calculated during the Monte Carlo simulations described in Sect. 2.2. When FS≤1,ofto be consistent with the creep equation,,deterministically$P_0$,$K_d$,diffusion coefficientfor the location of the landslide analysis based on publishedrates ofand diffusionCreation, calibration, and application of the soil evolution model are detailed in~~ Sect. 4.1.2.

2.5 Reproducibility

[revised manuscript text omitted]

**3.2.1. Vegetation and Soil parameters**

Vegetation classes are obtained from the NLCD,  in 30 m resolution (Jin, 2013; USGS, 2014b). Parameters of a triangular distribution for $C$, $\phi$, $T$, and $h_s$ are provided in Table 1. In our case study, $C$ represents root cohesion. Soils across the study domain are assumed  cohesionless, due to low clay content (<10%)  this mountain substrate with large clasts (Kulhawy et al., 1990). Estimating root cohesion is challenging because of temporal and spatial variability in root density and size, differential breakage or pullout mechanisms, interaction among roots, and difficulty in measuring at a field scale (Pollen and Simon 2005; Schwarz et al., 2013). We developed  spatial coverages for minimum, mode, and maximum C for NOCA by relating vegetation classes with corresponding published C values in the literature (Table 1), where field observations suggest right-skewed distribution (Hammond et al., 1992; Schmidt et al., 2001; Gabet and Dunne 2002; Hales  et al., 2013). Based on ranges available in the literature, we selected a mode value as a commonly reported value, minimum parameter as 30% of the mode, representing death and loss of productivity (Sidle, 1991; 1992), and a maximum near the highest reported value for C. Forest have higher C than shrubland because of the greater root area and deeper roots (Arnone et al., 2016b).  Small C values are assigned for barren and developed land uses (~14% of the domain) having minimal vegetation. Mode values of C mapped over NOCA are  shown in Fig. 4b. Forest communities of the valley bottom and lower valley walls show high values of C, which declines as vegetation transitions from forests to shrublands to herbaceous communities with increasing elevation.

**Table 1.** Parameters defined for vegetation and soil types in the study region. For spatially continuous variables T and $h_s$ obtained directly from SSURGO, values represent spatial  statistics .

[revised manuscript text omitted]

The maximum and minimum soil depth parameters of the triangular distribution  were obtained by analyzing soil evolution model results. At most θ, CA, and Curv triplets used, a landslide occurred at least once over the modeled duration.  As described in Sect. 3.2.1, given the negatively-skewed nature of the temporally evolved soil depth (Figure 6 a,c), the maximum  soil depth parameter of the triangular distribution was set equal to 10% of the mode in all model simulations. Two scenarios for the minimum parameter of the triangular distribution were used to reflect soil depth uncertainty for contemporary and long-term conditions.  In the first case, we set the minimum parameter as 70% of the mode. The LandslideProbability model was run for this scenarios for both SSURGO (SSURGO-SD) and modeled soil depth (M-SD) input. In the long-term scenario, the minimum soil depth was set to 0.005 m, reflecting bedrock scour conditions by landslides.  We argue that this assumption implicitly introduces a temporal uncertainty component to soil depth, which may be used to more accurately estimate landslide return

period over the long-term. The model run was called M-SD LT for this case. ~~Two scenarios for the minimum parameter of the triangular distribution were used M-SD. scenarios were developed to compare with SSURGO-SD In the first approach we focus on existing contemporary soil depth conditions in the field by running two simulations called SSURGO-SD and M-SD where we set the minimum parameter as 70% of the mode. Second, we aimed to reflect the longer-term perspective of soil evolution on the uncertainty of soil depth (called M-SD LT simulation) by setting the minimum soil depth to 0.005 m, reflecting bedrock scour conditions by landslides. This assumption implicitly introduces a temporal uncertainty component to soil depth, which can be used to more accurately estimate landslide return period over the long-term.~~

**4.2 Probability of Failure**

Modeled annual probability of failure of shallow landslides, P(F), for NOCA simulated by the Landlab LandslideProbability component using SSURGO-SD and two M-SD scenarios are shown in Fig. 8.  run model run  3,000 values were sampled (i.e., iterations) for model parameter s at each grid cell in the Monte Carlo simulations.

P(F) derived from simulations exhibit low probabilities where slopes are moderate and cohesion is high (e.g., forest). Highly unstable areas largely correspond to steep barren landscape (13% of the model domain) mostly located below retreating alpine glaciers, with steep glacial landforms, transitioning from glacier to colluvial processes (similar to Guthrie and Brown 2008; Tarolli et al., 2008; Legg et al., 2014) (Fig. 9). These areas with a thin veneer colluvium, except for moraines, appear to be "continuously sliding" (Borga et al., 2002) or "chronically unstable" (Montgomery, 2001). Frequent slides impede the colonization of vegetation (Dietrich et al., 1995; Istanbulluoglu and Bras, 2005). Slides in barren areas were not completely included in our landslide inventory as they do not pose major risks to humans and infrastructure.

s (i.e., 3,000 iterations in each simulation).

[Figure]

a) SSURGO-SD     b) M-SD     c) M-SD LT

NOCA

Debris Avalanches

**Probability of Failure**

| | | | | |
|---|---|---|---|---|
| < 0.005 | 0.01 to 0.02 | 0.04 to 0.1 | 0.25 to 0.5 | > 0.9 |
| 0.005 to 0.01 | 0.02 to 0.04 | 0.1 to 0.25 | 0.5 to 0.9 | |

d) SSURGO-SD     e) M-SD     f) M-SD LT

[revised manuscript text omitted]
). Total cohesion has been found to affect FS estimates more on thin soil than on thick soils (Hammond et al., 1992). The sensitivity of FS to cohesion is even more pronounced on steep slopes, especially when saturated (Sidle 1984). Forest vegetation has also been found to stabilize slopes through the hydrological process or

25   root water updake and transpiration, which leads to drier soil conditions (Arnone et al., 2016b). In aggregate, thinner soils at higher elevations lead to lower mean P(F), which we referred to as soil depth control (see also Sidle 1984). The general contribution of elevation on the spatial organization of P(F) is labeled in Fig 10b.

**Comment [EI4]:** What is this I would delte this where did this come from.. response to reviwer.. this read like a lit review.

[Figure]

[Figure]

[revised manuscript text omitted]
- Model  confirmation with limited observations revealed similar model confidence for the three hazard maps, suggesting suitable use as relative hazard products. Validation of the model with observed landslides is hindered by the completeness and accuracy of the inventory, estimation of source areas, and unmapped landslides.
- Our shallow landslide hazard model provides regional scale estimates of the relative annual probability of shallow landslide initiation as well as landslide return period,

which is useful for civil protection through land use planning to minimize geohazard consequences from precipitation triggers.

**6 Data and Model Availability**

To facilitate ease of use of the landslide hazard model, we developed the landslide model as a
5   component of Landlab, an open-source Python toolkit for two-dimensional numerical modeling of Earth-surface dynamics available at GitHub: http://github.com/landlab/landlab (Hobley et al., 2017). Documentation, installation instructions, and software dependencies for the entire Landlab project can be found at: http://landlab.github.io/. The Landlab project is tested on recent-generation Mac, Linux and Windows platforms using Python versions 2.7, 3.4, and 3.5.
10  The Landlab modeling framework is distributed under a MIT open-source license. A component user User manual Manual and driver scripts for the application of the Landlab LandsideProbability component can be found at https://github.com/RondaStrauch/pub_strauch_etal_esurfhttps://github.com/landlab/pub_str auch_etal_esurf (Strauch, GitHub Repository).

Online access to the Landlab LandslideProbability model is freely provided through https://www.hydroshare.org, where data and code drivers are available to demonstrate and explore the model using interactive IPython notebooks in a JupyterHub. Thus, users can access, test, adapt, and apply the landslide model for their area of interest without downloading
20  Landlab or the components. Data and driver code used in this analysis are available at hydroshare (Strauch et al., 2017). Existing demonstration driver codes can be adapted to fit data provided in raster format by the user to create distributed data fields used as parametervariables in the component. Instructions for accessing HydroShare and the online demonstrations, codes, and data used in this paper are provided in the Ssupplemental material.

**Acknowledgements**

This research was supported by the US National Science Foundation (CBET-1336725, OAC-1450412, 1450409, 1450338) and USGS Northwest Climate Science Center. We thank Dan Miller and two anonymous reviewers for helpful reviews of an earlier version of the
30  manuscript. Technical editing on portions of the manuscript was provided by Brad Strauch and Diann Strom. We also appreciate the developers of Landlab, including open-source contributors to the earth surface processes modeling community. Data repository (Strauch et al., 2017), testing, and online reproducibility was facilitated by the cyber infrastructure of HydroShare services provided by researchers associated with the Consortium of Universities for
35  the Advancement of Hydrologic Science, Inc. (CUAHSI), particularly the support of Tony Castronova, and the team at CyberGIS Center for Advanced Digital and Spatial Studies for their maintenance and support for our use of the ROGER Supercomputer at the National Center for Supercomputing Applications (NCSA) at University of Illinois at Urbana-Champaign.

[revised manuscript text omitted]

Font: Cambria Math, Highlight

| Page 13: [1] Formatted | Erkan Istanbulluoglu | 09/10/2017 21:04:00 |
|---|---|---|

Font: Cambria Math, Highlight

| Page 13: [1] Formatted | Erkan Istanbulluoglu | 09/10/2017 21:04:00 |
|---|---|---|

Font: Cambria Math, Highlight

| Page 13: [1] Formatted | Erkan Istanbulluoglu | 09/10/2017 21:04:00 |
|---|---|---|

Font: Cambria Math, Highlight

| Page 13: [1] Formatted | Erkan Istanbulluoglu | 09/10/2017 21:04:00 |
|---|---|---|

Font: Cambria Math, Highlight

| Page 13: [1] Formatted | Erkan Istanbulluoglu | 09/10/2017 21:04:00 |
|---|---|---|

Font: Cambria Math, Highlight

| Page 13: [1] Formatted | Erkan Istanbulluoglu | 09/10/2017 21:04:00 |
|---|---|---|

Font: Cambria Math, Highlight

| Page 13: [1] Formatted | Erkan Istanbulluoglu | 09/10/2017 21:04:00 |
|---|---|---|

Font: Cambria Math, Highlight

| Page 13: [1] Formatted | Erkan Istanbulluoglu | 09/10/2017 21:04:00 |
|---|---|---|

Font: Cambria Math, Highlight

| Page 13: [1] Formatted | Erkan Istanbulluoglu | 09/10/2017 21:04:00 |
|---|---|---|

Font: Cambria Math, Highlight

| Page 13: [1] Formatted | Erkan Istanbulluoglu | 09/10/2017 21:04:00 |
|---|---|---|

Font: Cambria Math, Highlight

| Page 13: [1] Formatted | Erkan Istanbulluoglu | 09/10/2017 21:04:00 |
|---|---|---|

Font: Cambria Math, Highlight

| Page 13: [1] Formatted | Erkan Istanbulluoglu | 09/10/2017 21:04:00 |
|---|---|---|

Font: Cambria Math, Highlight

| Page 13: [2] Formatted | Erkan Istanbulluoglu | 09/10/2017 21:04:00 |
|---|---|---|

Font: Cambria Math, Highlight

| Page 13: [2] Formatted | Erkan Istanbulluoglu | 09/10/2017 21:04:00 |
|---|---|---|

Font: Cambria Math, Highlight

| Page 13: [2] Formatted | Erkan Istanbulluoglu | 09/10/2017 21:04:00 |
|---|---|---|

Font: Cambria Math, Highlight

| Page 13: [2] Formatted | Erkan Istanbulluoglu | 09/10/2017 21:04:00 |
|---|---|---|

Font: Cambria Math, Highlight

| Page 13: [2] Formatted | Erkan Istanbulluoglu | 09/10/2017 21:04:00 |
|---|---|---|

Font: Cambria Math, Highlight

| Page 13: [2] Formatted | Erkan Istanbulluoglu | 09/10/2017 21:04:00 |
|---|---|---|

Font: Cambria Math, Highlight

| Page 13: [2] Formatted | Erkan Istanbulluoglu | 09/10/2017 21:04:00 |
|---|---|---|

Font: Cambria Math, Highlight

| Page 13: [2] Formatted | Erkan Istanbulluoglu | 09/10/2017 21:04:00 |

Font: Cambria Math, Highlight

| Page 13: [3] Formatted | Erkan Istanbulluoglu | 09/10/2017 21:04:00 |

Highlight

| Page 13: [3] Formatted | Erkan Istanbulluoglu | 09/10/2017 21:04:00 |

Highlight

| Page 13: [3] Formatted | Erkan Istanbulluoglu | 09/10/2017 21:04:00 |

Highlight

| Page 13: [3] Formatted | Erkan Istanbulluoglu | 09/10/2017 21:04:00 |

Highlight

| Page 13: [3] Formatted | Erkan Istanbulluoglu | 09/10/2017 21:04:00 |

Highlight

| Page 13: [4] Formatted | Erkan Istanbulluoglu | 09/10/2017 21:04:00 |

Font: Cambria Math, Highlight

| Page 13: [4] Formatted | Erkan Istanbulluoglu | 09/10/2017 21:04:00 |

Font: Cambria Math, Highlight

| Page 13: [4] Formatted | Erkan Istanbulluoglu | 09/10/2017 21:04:00 |

Font: Cambria Math, Highlight

| Page 13: [4] Formatted | Erkan Istanbulluoglu | 09/10/2017 21:04:00 |

Font: Cambria Math, Highlight

| Page 13: [4] Formatted | Erkan Istanbulluoglu | 09/10/2017 21:04:00 |

Font: Cambria Math, Highlight

| Page 13: [4] Formatted | Erkan Istanbulluoglu | 09/10/2017 21:04:00 |

Font: Cambria Math, Highlight

| Page 13: [4] Formatted | Erkan Istanbulluoglu | 09/10/2017 21:04:00 |

Font: Cambria Math, Highlight

| Page 13: [4] Formatted | Erkan Istanbulluoglu | 09/10/2017 21:04:00 |

Font: Cambria Math, Highlight

| Page 13: [4] Formatted | Erkan Istanbulluoglu | 09/10/2017 21:04:00 |

Font: Cambria Math, Highlight

| Page 13: [4] Formatted | Erkan Istanbulluoglu | 09/10/2017 21:04:00 |

Font: Cambria Math, Highlight

| Page 13: [4] Formatted | Erkan Istanbulluoglu | 09/10/2017 21:04:00 |

Font: Cambria Math, Highlight

| Page 13: [4] Formatted | Erkan Istanbulluoglu | 09/10/2017 21:04:00 |

Font: Cambria Math, Highlight

| Page 13: [4] Formatted | Erkan Istanbulluoglu | 09/10/2017 21:04:00 |

Font: Cambria Math, Highlight

| Page 13: [4] Formatted | Erkan Istanbulluoglu | 09/10/2017 21:04:00 |

Font: Cambria Math, Highlight

| Page 13: [4] Formatted | Erkan Istanbulluoglu | 09/10/2017 21:04:00 |

Font: Cambria Math, Highlight

| Page 13: [4] Formatted | Erkan Istanbulluoglu | 09/10/2017 21:04:00 |

Font: Cambria Math, Highlight

| Page 13: [4] Formatted | Erkan Istanbulluoglu | 09/10/2017 21:04:00 |

Font: Cambria Math, Highlight

| Page 13: [4] Formatted | Erkan Istanbulluoglu | 09/10/2017 21:04:00 |

Font: Cambria Math, Highlight

| Page 13: [4] Formatted | Erkan Istanbulluoglu | 09/10/2017 21:04:00 |

Font: Cambria Math, Highlight

| Page 13: [4] Formatted | Erkan Istanbulluoglu | 09/10/2017 21:04:00 |

Font: Cambria Math, Highlight

| Page 13: [5] Formatted | Ronda Strauch | 29/09/2017 16:10:00 |

Font: Not Italic

| Page 13: [5] Formatted | Ronda Strauch | 29/09/2017 16:10:00 |

Font: Not Italic

| Page 13: [5] Formatted | Ronda Strauch | 29/09/2017 16:10:00 |

Font: Not Italic

| Page 13: [5] Formatted | Ronda Strauch | 29/09/2017 16:10:00 |

Font: Not Italic